# FraGNNet: A Deep Probabilistic Model for Tandem Mass Spectrum Prediction

**Adamo Young**                                        *ayoung@cs.toronto.edu*
*Department of Computer Science, University of Toronto*
*Vector Institute for Artificial Intelligence*
*Terrence Donnelly Centre for Cellular and Biomolecular Research, University of Toronto*

**Fei Wang**                                        *fw4@ualberta.ca*
*Department of Computing Science, University of Alberta*
*Alberta Machine Intelligence Institute*

**David S Wishart**                                        *dwishart@ualberta.ca*
*Department of Computing Science, University of Alberta*
*Department of Biological Sciences, University of Alberta*
*Department of Laboratory Medicine and Pathology, University of Alberta*
*Faculty of Pharmacy and Pharmaceutical Sciences, University of Alberta*

**Bo Wang**                                        *bowang@vectorinstitute.ai*
*Department of Computer Science, University of Toronto*
*Vector Institute for Artificial Intelligence*
*Department of Laboratory Medicine and Pathobiology, University of Toronto*
*Peter Munk Cardiac Centre, University Health Network*

**Russell Greiner**                                        *rgreiner@ualberta.ca*
*Department of Computing Science, University of Alberta*
*Alberta Machine Intelligence Institute*

**Hannes Rost**                                        *hannes.rost@utoronto.ca*
*Department of Molecular Genetics, University of Toronto*
*Terrence Donnelly Centre for Cellular and Biomolecular Research, University of Toronto*

**Reviewed on OpenReview:** *https://openreview.net/forum?id=UsqeHx9Mbx*

## Abstract

Compound identification from tandem mass spectrometry (MS/MS) data is a critical step in the analysis of complex mixtures. Typical solutions for the MS/MS spectrum to compound (MS2C) problem involve comparing the unknown spectrum against a library of known spectrum-molecule pairs, an approach that is limited by incomplete library coverage. Compound to MS/MS spectrum (C2MS) models can improve retrieval rates by augmenting real libraries with predicted MS/MS spectra. Unfortunately, many existing C2MS models suffer from problems with mass accuracy, generalization, or interpretability. We develop a new probabilistic method for C2MS prediction, FraGNNet, that can efficiently and accurately simulate MS/MS spectra with high mass accuracy. Our approach formulates the C2MS problem as learning a distribution over molecule fragments. FraGNNet achieves state-of-the-art performance in terms of prediction error and surpasses existing C2MS models as a tool for retrieval-based MS2C.

# 1 Introduction

Small molecule identification is a challenging problem with broad scientific implications. Determining the chemical composition of a liquid sample (such as human blood or plant extract) is a critical step both in the discovery of novel compounds and in the recognition of known compounds in new contexts. Tandem mass spectrometry (MS/MS) is a widely employed tool for molecule identification, with applications ranging from drug discovery to environmental science and metabolomics (Dueñas et al., 2022; Gowda & Djukovic, 2014; Peters, 2011; Lebedev, 2013). Analyzing the set of MS/MS spectra measured from a liquid sample can reveal important information about its constitution: each spectrum acts as a chemical signature that can help identify a molecule in the sample.

The MS/MS spectrum to compound (MS2C) problem is the task of inferring the structure of a molecule from its mass spectrum. Existing MS2C workflows often rely on comparing the spectrum of an unknown molecule against a library of reference MS/MS spectra with known identities. However, spectral libraries are far from comprehensive. The largest public MS/MS library, NIST23 (Stein, 2023), consists of 51,501 compounds and 2.4 million spectra. In contrast, the most commonly used chemical database, PubChem (Kim et al., 2019), contains 119 million compounds at the time of writing. The huge gap in spectrum coverage means *retrieval-based* MS2C might not work even for molecules that have previously been observed.

Many computational approaches attack the MS2C problem head-on by attempting to predict information about the molecule from the MS/MS spectrum. Some models infer high-level chemical properties from the spectrum (Dührkop et al., 2015; Voronov et al., 2022) and use these features to recommend likely candidate structures from existing chemical databases (Dührkop et al., 2015; Dührkop et al., 2019; Goldman et al., 2023b;c; Bushuiev et al., 2025) or guide generative models towards potential matches (Stravs et al., 2022). Other methods (Butler et al., 2023; Shrivastava et al., 2021; Wang et al., 2025b; Bohde et al., 2025) directly generate structures from the spectrum.

Compound to mass spectrum (C2MS) prediction can be viewed as an indirect approach for solving the MS2C problem. Instead of attempting to infer structural features from the MS/MS spectrum, C2MS models work by boosting the effectiveness of existing retrieval-based MS2C workflows. Accurate C2MS models can predict MS/MS spectra for millions of molecules that are missing from spectral libraries, improving coverage by orders of magnitude. This kind of library augmentation has been a mainstay in retrieval-based MS2C workflows for over a decade (Wolf et al., 2010; Allen et al., 2015; Wishart et al., 2018) and has led to the successful identification of many novel compounds (Skinnider et al., 2021; Wang et al., 2023; Qiang et al., 2024).

Despite numerous advances, challenges with C2MS remain (Schymanski et al., 2017; Bushuiev et al., 2024). Many existing approaches cannot predict MS/MS spectra with high mass accuracy; this loss of information can be harmful in MS2C applications (Kind et al., 2018). Model interpretability is important for manual validation of MS/MS predictions, yet many models operate as black boxes. Finally, generalization to new compounds is difficult given the limited availability of training data.

In this work, we make the following contributions:

- We introduce FraGNNet, a novel method for MS/MS spectrum prediction that integrates combinatorial bond-breaking methods with principled probabilistic modelling.

- We argue that this approach meets key requirements in the areas of prediction error, mass accuracy and interpretability.

- Through comparisons with strong baseline models, we demonstrate that FraGNNet achieves state-of-the-art performance on MS/MS spectrum prediction and compound retrieval tasks.

# 2 Background

At a high level, MS/MS spectrometry provides information about a molecule's structure by measuring how it breaks down. The experimental process is outlined in Figure 1. First, molecules in the sample are ionized: at this stage they are referred to as *precursor ions*, because they have not yet undergone fragmentation. In liquid

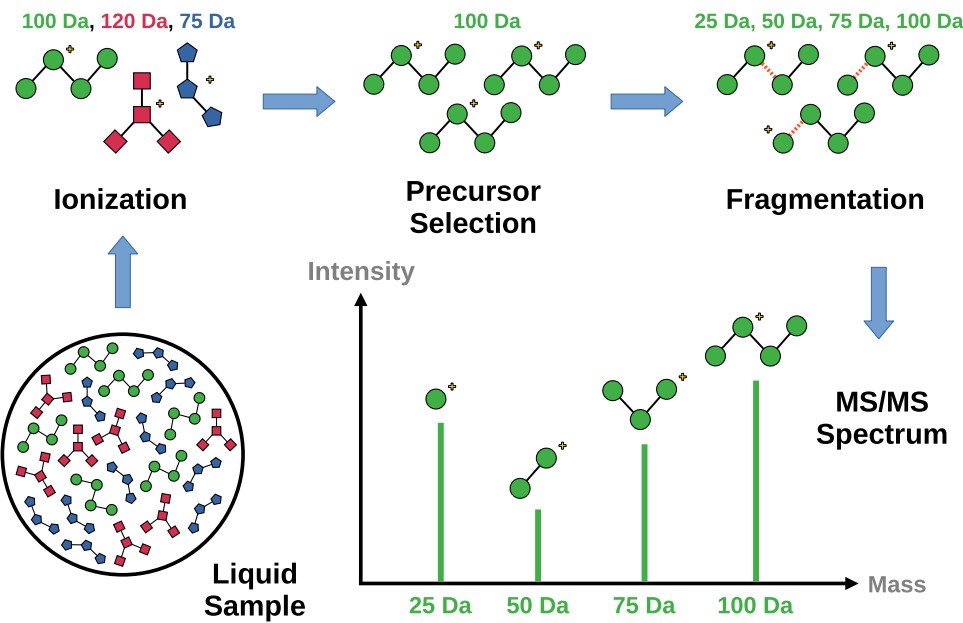

Figure 1: Overview of MS/MS spectrometry: molecules in the sample are ionized to form precursors, filtered by precursor mass (100 Da), and sent for fragmentation. The fragmentation process stochastically produces fragments with mass values of 25, 50, 75 Da. The distribution of precursor and fragment mass values forms the spectrum.

samples, each molecule may become associated with a charged *adduct* during ionization. For example, if the adduct is a hydrogen ion $H^+$, an $[M+H]^+$ precursor ion will form. Following ionization, the mass to charge ratio $(m/z)$ of each precursor ion is subsequently measured using a mass analyzer. Once the precursor $m/z$ values have been measured, ions of a selected $m/z$ are isolated and subjected to fragmentation for further analysis. The fragmentation process is typically facilitated by energetic collisions with neutral gas particles, and is influenced by experimental parameters such as *collision energy*. In general, higher collision energy results in more extensive fragmentation.

A *fragment* is defined as a molecule composed of a subset of the atoms in a precursor molecule. The fragmentation process involves chemical reactions that cause the stochastic breakage and formation of bonds in the molecule. After fragmentation, the resulting fragment ions are sent to a mass analyzer for $m/z$ measurement, producing a distribution over $m/z$ values that is called the MS/MS spectrum. It is possible for different fragments to have nearly identical $m/z$, complicating analysis of the spectrum. Throughout this work, we focus our analysis on $[M+H]^+$ spectra: as $z$ is always $+1$, the spectrum can be directly interpreted as a distribution over masses. However, our methods can be extended to other types of adducts, such as $[M-H]^-$ and $[M+Na]^+$.

Mathematically, an MS/MS spectrum $Y$ can be represented as a finite set of pairs $\{(m_j, P(m_j))\}_j$ where each mass $m_j$ has an associated probability $P(m_j)$. We refer to each individual pair $(m_j, P(m_j))$ as a *peak*, where the mass $m_j$ (measured in Daltons or Da) is called the peak *location* and the probability $P(m_j)$ is called the peak *intensity*. The precision of the peak locations is commonly referred to as the *mass accuracy*. The C2MS problem can be formulated as a standard supervised learning task. The dataset is composed of $N$ tuples $\{(X_i, Y_i)\}_{i=1}^N$ where $X_i$ is a molecule and $Y_i$ is its associated MS/MS spectrum. The goal of a C2MS model is to predict $Y_i$ from $X_i$. The set of masses in a particular spectrum $Y_i$ is denoted as $M_i$.

Every molecule can be represented as an undirected *molecular graph* $G = (V, E)$. Each node $a \in V$ represents an atom in the molecule with an associated element label $\omega_a \in \Omega$ (where $\Omega = \{C, H, N, O, P, S, \ldots\}$ is a finite set of common elements) and each edge $b \in E$ represents a covalent bond between atoms. A *molecular formula* is a representation of the molecule that captures only the quantity of each type of atom present in

the graph. For example, the molecule methylaminomethanol (Figure 5) has a molecular formula of $C_2H_7NO$, which means that its molecular graph contains two carbon atoms, seven hydrogens, one nitrogen, and one oxygen.

*Peak annotation* is the process of linking peaks in an MS/MS spectrum to chemical information. Practitioners often use these annotations to interpret the spectrum: in the context of MS2C, these annotations can help users compare possible candidate molecules by facilitating the incorporation of domain knowledge (Steckel & Schlosser, 2019; Johnson & Carlson, 2015). When a peak is associated with a subformula of the precursor molecular formula, this is called a *formula annotation*; similarly, when a peak is associated with a fragment of the precursor molecule this is called a *fragment annotation*.

## 3 Related Work

The increasing availability of small molecule MS/MS data has lead to a proliferation of machine learning models for C2MS prediction. Existing methods can be broadly grouped into two categories, *binned* and *structured*, based on how they represent the spectrum.

Binned methods approximate the spectrum by discretizing the mass range into a fixed number of equally sized bins, each with an associated intensity. This simplifies the spectrum prediction problem to a vector regression task, which can be readily solved without extensive domain-specific model customization. Binned approaches generally vary based on the strategy they employ for encoding the input molecule: multi-layer perceptrons (Wei et al., 2019), 2D and 3D graph neural networks (Zhu et al., 2020; Li et al., 2022; Hong et al., 2023; Park et al., 2024), and graph transformers (Young et al., 2023) have all been used successfully. However, selecting an appropriate bin size can be challenging: bins that are too large result in loss of information, while bins that are too small can be overly sensitive to measurement error and yield high-dimensional output vectors.

Structured approaches sidestep the binning problem by modelling the spectrum as a distribution over chemical formulae. This representation allows for arbitrarily high mass accuracy, since the formula masses can be used to determine peaks locations with precision. Some methods predict the formula distribution directly, using either autoregressive formula generation (Goldman et al., 2023a) or a large fixed formula vocabulary (Murphy et al., 2023). Others rely on recursive fragmentation (Wang et al., 2021; Zhu & Jonas, 2023; Nowatzky et al., 2025) or autoregressive generation (Goldman et al., 2024; Wang et al., 2025a) to first model a distribution over fragments, then map this to a distribution over formulae. Structured C2MS models tend to be more interpretable, since each peak is always associated with a formula (and possibly one or more fragments). However, they can also be slower than binned approaches due to the increased complexity of the spectrum representation.

CFM-ID (Allen et al., 2015; Wang et al., 2021) and ICEBERG (Goldman et al., 2024) are structured C2MS models that explicitly model fragmentation. Like FraGNNet, both approaches algorithmically generate fragments of the input molecule and predict a distribution over those fragments. Their principal distinction, however, lies in the specific strategy for fragment generation. CFM-ID employs a deterministic bond-breaking algorithm and incorporates chemistry domain knowledge to compute a set of plausible fragments. This approach benefits from consistent fragment generation and strong chemical priors; however, it does not scale well to larger molecules and datasets. In contrast, ICEBERG uses a deep learning model to autoregressively generate fragments. This approach has proven to be overall more accurate and scalable than CFM-ID (Goldman et al., 2024), but it relies on a learned fragment generation process that can result in errors. FraGNNet takes a pragmatic approach: like CFM-ID, it employs a deterministic algorithm to generate a comprehensive set of fragments; however, by relaxing chemical constraints it can achieve greater efficiency and scale to larger compounds like ICEBERG.

## 4 Methods

### 4.1 Overview

The goal of our method is to predict the MS/MS spectrum for an input molecular structure. The model works in two stages: first, a recursive bond-breaking algorithm (Section 4.2) generates a set of plausible

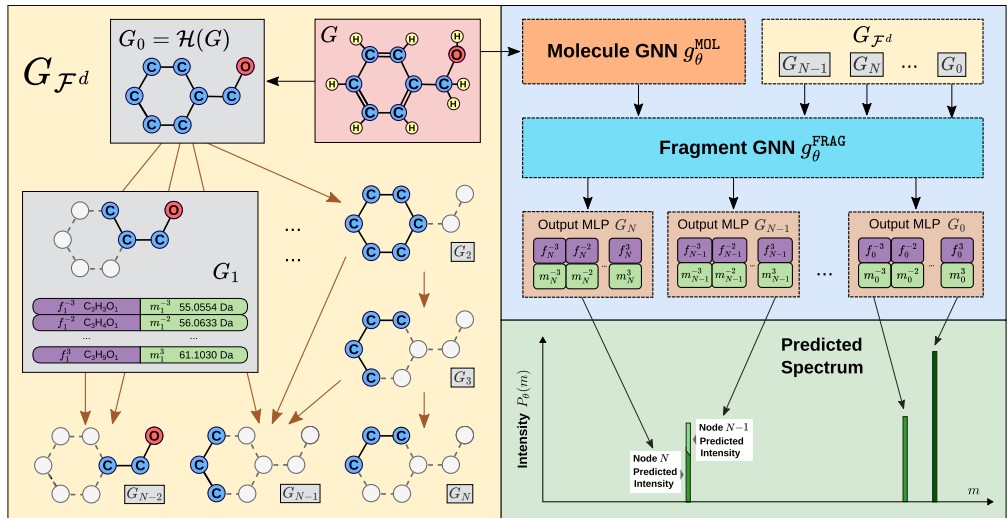

Figure 2: Overview of the FraGNNet C2MS model. The input molecule ($G$, red box) is processed into an approximate Fragmentation DAG ($G_{\mathcal{F}^d}$, yellow box) and independently embedded by the Molecule GNN ($g_\theta^{\texttt{MOL}}$, orange box). Information from the DAG is combined with the molecule embedding and processed by the Fragment GNN ($g_\theta^{\texttt{FRAG}}$, blue box). Output MLPs (brown boxes) are applied to each fragment node $n$ in the DAG to predict a per-node distribution over formulae (purple boxes), which can be mapped to a distribution over masses (green boxes) and summed across nodes to create the MS/MS spectrum.

molecule fragments. Then, a probabilistic model (Section 4.3) parameterized by a graph neural network (Section 4.4) predicts a distribution over these fragments. The fragment distribution induces a distribution over molecular formulae, which is converted to an MS/MS spectrum using formula masses (Section 4.3). This approach allows for extremely high mass accuracy and probabilistic formula and fragment annotations.

## 4.2 Recursive Fragmentation

Recall our definition of the molecular graph $G = (V, E)$ from Section 2, where nodes $a \in V$ represents atoms and edges $b \in E$ represent bonds. Let $S(G)$ be the set of connected subgraphs of the graph $G$.

**Definition 4.1.** The graph $\mathcal{F}(G) = (\mathcal{F}_1(G), \mathcal{F}_2(G))$ is called a fragmentation DAG with respect to the graph $G$ if the following properties hold:

1. Each node $n \in \mathcal{F}_1(G)$ corresponds to a connected subgraph $G_n \in S(G)$

2. Each edge $e \in \mathcal{F}_2(G)$ from node $u \in \mathcal{F}_1(G)$ to node $v \in \mathcal{F}_1(G)$ exists if and only if $G_v \in S(G)$ is a connected subgraph of $G_u \in S(G)$ that can be constructed by removing a single edge from $G_u$ and selecting one of the resulting connected subgraphs.

Note that the root node $r$ of $\mathcal{F}(G)$ always corresponds to the original graph $G$, and the leaves of $\mathcal{F}(G)$ correspond to individual atoms $a \in V$.

Our method assumes that most fragments that contribute to an MS/MS spectrum can be modelled as products of a sequence of bond breakages. Such fragments appear as nodes $n \in \mathcal{F}_1(G)$, and their fragmentation history can be represented as a path from the root $r$ to $n$. We can derive a set of molecular formulae $\{f_n : n \in \mathcal{F}_1(G)\}$, where each $f_n$ represents the molecular formula of $G_n$. This set can be used to calculate the set of possible peak locations in the spectrum, $M(G) = \{m_n : n \in \mathcal{F}_1(G)\}$ where $m_n = \text{mass}(f_n)$ is the monoisotopic mass of molecular formula $f_n$.

The fragmentation DAG $\mathcal{F}(G)$ can be a useful tool for spectrum prediction (Wang et al., 2021; Goldman et al., 2023a; 2024), providing information about peak locations and relationships between fragments in the

spectrum. However, computing $\mathcal{F}(G)$ from $G$ through exhaustive edge removal is expensive. Inspired by previous approaches from the literature (Wolf et al., 2010; Ruttkies et al., 2016; Allen et al., 2015; Ridder et al., 2014; Goldman et al., 2024), we approximate $\mathcal{F}(G)$ using a few simplifying assumptions.

**Definition 4.2.** The graph $\mathcal{H}(G) = (\mathcal{H}_1(G), \mathcal{H}_2(G))$ is called a heavy atom skeleton of $G$ if $\mathcal{H}(G)$ is the largest connected subgraph of $G$ such that $\omega_a \neq H$ (hydrogen) for all $a \in \mathcal{H}_1(G)$.

Since $|\mathcal{H}_2(G)|$ is often smaller than $|E|$ (in the NIST20 MS/MS dataset, $\approx 43\%$ smaller on average), calculating $\mathcal{F}(\mathcal{H}(G))$ is considerably faster than calculating $\mathcal{F}(G)$. We employ a recursive edge removal (*i.e.*, bond-breaking) algorithm that only considers nodes that are at most $d$ hops away from the root $r$, producing a connected subgraph $\mathcal{F}^d(\mathcal{H}(G))$ of $\mathcal{F}(\mathcal{H}(G))$ (see Algorithm 1 for an outline of our implementation).

For simplicity of notation, we refer to $\mathcal{F}^d(\mathcal{H}(G))$ as $G_{\mathcal{F}^d}$, with vertex set $V_{\mathcal{F}^d}$ and edge set $E_{\mathcal{F}^d}$. By definition, each node $n \in V_{\mathcal{F}^d}$ is associated with a fragment subgraph $G_n \in S(\mathcal{H}(G))$ that does not contain any hydrogen atoms. Since real fragments often include hydrogens, we employ heuristics to bound the number of hydrogens associated with each $G_n$. This approach enables us to account for hydrogens while avoiding the computational cost of explicitly modeling their positions in the graph.

Let $h_n$ be the number of hydrogen atoms in original molecular graph $G$ that are connected to an atom in the fragment subgraph $G_n$. We define the set $\{h_n - j, \ldots, h_n + j\}$ as the range of hydrogen counts for the subgraph $G_n$, where $j$ is an (integer) tolerance parameter. This induces a set of possible molecular formulae $\{f_n^{-j}, \ldots, f_n^j\}$ and associated masses $\{m_n^{-j}, \ldots, m_n^j\}$, where $f_n^i$ is $f_n$ with the addition of $h_n + i$ hydrogens, and $m_n^i$ is its corresponding monoisotopic mass. For example, if $f_n = C_2O_1$, $h_n = 3$, and $i = 2$, then $m_n^i = \text{mass}(C_2O_1H_5) = 45.03404$ Da.

**Definition 4.3.** The set

$$\hat{M}(G, d, j) \quad = \quad \bigcup_{n \in V_{\mathcal{F}^d}} \{m_n^{-j}, \ldots, m_n^j\}$$

is the set of masses derived from the approximate heavy-atom fragmentation DAG $G_{\mathcal{F}^d}$ with hydrogen tolerance $j$.

Empirically, we find that calculating $\hat{M}(G, d, j)$ (Definition 4.3) with $d \in \{3, 4\}$ and $j = 4$ can effectively capture most of the total peak intensity in real MS/MS data (see Table 4).

### 4.3 Probabilistic Formulation

Our model can be interpreted as a hierarchical latent variable model, whose latent distributions depend on a molecular graph $G$ and its approximate fragmentation DAG $G_{\mathcal{F}^d}$.

To begin, we define the following latent probability distributions:

**Definition 4.4.** Let $P_\theta(n)$ be a discrete finite probability distribution over the DAG nodes $N = V_{\mathcal{F}^d}$, parameterized by a neural network $g_\theta$.

**Definition 4.5.** Let $P_\theta(f|n)$ be a discrete finite conditional probability distribution between DAG nodes $N$ and associated formulae $F = \bigcup_{n \in N} \{f_n^{-j}, \ldots, f_n^j\}$, parameterized by a neural network $g_\theta$.

Both distributions depend implicitly on the molecular graph $G$, but for clarity of notation we have omitted this. Note that for each node $n$, $P_\theta(f|n)$ has support over $2j + 1$ formulae.[1]

The joint distribution $P_\theta(n, f) = P_\theta(n) P_\theta(f|n)$ can be loosely interpreted as identifying which substructures are generated during fragmentation, with $P_\theta(n)$ modelling the heavy atom structures of the probable fragments and $P_\theta(f|n)$ modelling the number of hydrogens associated with each of those fragments.

By marginalizing $P_\theta(n, f)$ over the nodes $n$, it is possible to calculate a distribution over molecular formulae $P_\theta(f)$. Since each formula $f$ has an associated mass, the discrete distribution $P_\theta(f)$ can be easily converted to a continuous distribution $P_\theta(m)$ over masses. Following Allen et al. (2015), we formulate $P_\theta(m)$ as a mixture of truncated univariate Gaussians as outlined in Equation 1:

---

[1] There is additional filtering for chemical validity which may remove some formulae, but for clarity this has been omitted.

$$P_\theta(m) \;=\; \sum_f P_\theta(f) \; P(m|f) \tag{1}$$

The conditional $P(m|f)$ is a narrow truncated Gaussian centered on the formula mass, $\mu(f) = \text{mass}(f)$, with variance $\sigma(f)$ proportional to $f$ and truncation occurring at $\pm 1$ standard deviation from the mean. This Gaussian model approximates the error distribution of the mass analyzer (Allen et al., 2015). At inference time it is convenient to approximate $P(m)$ as a discrete distribution with $P(m|f) = \delta(\text{mass}(f))$, where $\delta$ is the Dirac delta function.

Using Bayes Theorem, we can calculate another latent distribution $P_\theta(n|f)$ that identifies how much each fragment $n$ is contributing to a predicted peak centered at formula $f$. We use $P_\theta(n|f)$ to predict fragment annotations for each output peak (see Figure 3 and Section 5.4).

In Appendix A.3, we describe analogs of $P_\theta(n)$, $P_\theta(f|n)$, and $P_\theta(n|f)$ that account for fragment subgraph isomorphism.

### 4.4 Neural Network Parameterization

The distributions $P_\theta(n)$ and $P_\theta(f|n)$ are parameterized by a two-stage graph neural network (GNN, Battaglia et al. 2018) as defined by Equation 2:

$$g_\theta(G, G_{\mathcal{F}^d}) \;=\; g_\theta^{\texttt{FRAG}}\left( g_\theta^{\texttt{MOL}}(G), \; G_{\mathcal{F}^d} \right) \tag{2}$$

The first stage $g_\theta^{\texttt{MOL}}$, called the *Molecule GNN*, operates on the input molecular graph $G$. The second stage $g_\theta^{\texttt{FRAG}}$, called the *Fragment GNN*, combines information from the fragmentation DAG and the molecule embeddings to predict the distributions $P_\theta(n)$ and $P_\theta(f|n)$.

#### 4.4.1 Molecule GNN

The Molecule GNN $g_\theta^{\texttt{MOL}}$ takes the input molecular graph $G$ and outputs embeddings for the atoms in the graph. The atom embeddings $\bar{h}_a^{(0)}$ and bond embeddings $\bar{h}_b^{(0)}$ are initialized with specific features from $G$ (see Appendix A.4).

GNN models work by iteratively updating node states through the aggregation of neighbourhood information. $g_\theta^{\texttt{MOL}}$ uses the GINE architecture (Xu et al., 2019; Fey & Lenssen, 2019), which incorporates both node and edge information in its updates. The GINE update rule is given by Equation 3, where $l$ is the GNN layer index, $l \in \{1, \ldots, L_1\}$, and $q_\theta$ is a standard multi-layer perceptron (MLP):

$$\bar{h}_a^{(l+1)} \;=\; q_\theta\left( \bar{h}_a^{(l)} + \sum_{u \in \mathbb{N}(a)} \text{ReLU}(\bar{h}_u^{(l)} + \bar{h}_b^{(l)}) \right) \tag{3}$$

The final atom embeddings $\bar{h}_a^{(L_1)}$ are subsequently passed to the Fragment GNN $g_\theta^{\texttt{FRAG}}$ for further processing.

#### 4.4.2 Fragment GNN

The Fragment GNN $g_\theta^{\texttt{FRAG}}$ is another GINE network that propagates information along the approximate fragmentation DAG. Each DAG node $n \in V_{\mathcal{F}^d}$ is featurized using information about its associated subgraph $G_n$, precisely described in Equations 4 and 5. The vector $\bar{h}_n^{(0)}$ is a concatenation of three terms: $\hat{h}_n^s$ is the average atom embedding for atoms in $G_n$; $\hat{h}_n^f$ is an embedding of the subgraph formula $f_n$; and $\hat{h}_n^d$ is an embedding of the depth in the DAG at which node $n$ is located.

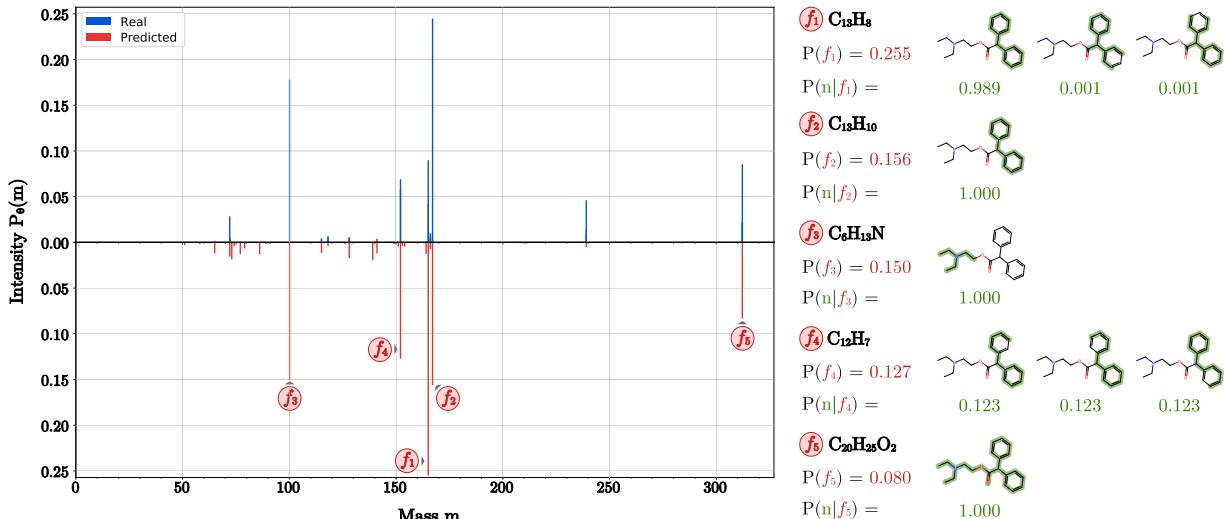

Figure 3: Example Spectrum Prediction with Peak Annotations: A real spectrum (blue) for a molecule from the test set, Adiphenine, is compared with its corresponding FraGNNet-D4 prediction (red). The five most intense peak predictions are annotated with formulae and up to three probable fragment structures. The fragment annotation for peak 5 clearly corresponds to the precursor ion (*i.e.*, the entire graph). The model is uncertain about the fragment annotation for peak 4, giving equal probability (0.123) to three related structures and significant probability (0.631) to other less likely structures.

$$\bar{h}_n^{(0)} \;=\; \hat{h}_n^s \;\|\; \hat{h}_n^f \;\|\; \hat{h}_n^d \tag{4}$$

$$\hat{h}_n^s \;=\; \frac{1}{|V_n|} \sum_{a \in V_n} \bar{h}_a^{(L_1)} \tag{5}$$

The edge embeddings $\bar{h}_e^{(0)}$ are also initialized with subgraph information: they capture differences between adjacent DAG nodes. Refer to Appendix A.5 for full details.

The DAG nodes are processed by the Fragment GNN in a manner that is similar to Equation 3. After $L_2$ layers of processing, a small output MLP is applied to each node embedding $\bar{h}_n^{(L_2)}$, producing a $2j + 1$ dimensional vector representing the unnormalized logits for $P_\theta(n, f_n^i)$ for each $i \in \{-j, \ldots, j\}$. The other latent distributions in Section 4.3 are calculated from the joint through normalization, marginalization, and application of Bayes Theorem.

After performing model ablations (see Appendix A.8), we discovered that edge information could be omitted without degradation in performance, allowing for faster training and inference. As a result, the experiments in Section 5 used a variant of $g_\theta^{FRAG}$ that excludes the neighbourhood aggregation term from the GINE update, effectively acting as a node-wise MLP.

## 4.5 Loss Function

We fit the parameters of the model $\theta$ with maximum likelihood estimation, using stochastic gradient descent. The loss function is based on the negative log-likelihood of the data, defined in Equation 6:

$$\mathcal{L}_{\text{NLL}}(\theta) \;=\; \frac{1}{I} \sum_{i=1}^{I} \sum_{m \in M_i} -P(m) \log P_\theta(m) \tag{6}$$

For each spectrum (indexed by $i$), a subset of the peak masses $M_i^{\text{OS}} \subseteq M_i$ are defined to be *Outside of the Support (OS)* if they are far enough from away the predicted set of masses $\hat{M}(G_i, d, j)$ such that their predicted probability is 0 (Equation 7).

$$P(M_i^{\text{OS}}) = \sum_{m \in M_i : P_\theta(m)=0} P(m) \tag{7}$$

The rest of the masses $M_i^{\text{IS}} = M_i - M_i^{OS}$ are deemed to be *Inside of the Support (IS)*.

Modelling the OS probability can provide useful information about the reliability of the predicted spectrum (Appendix A.7). FraGNNet is trained to predict $P_\theta(M_i^{\text{OS}})$, an estimation of $P(M_i^{\text{OS}})$. Adjusting the loss function $\mathcal{L}_{\text{NLL}}(\theta)$ to incorporate the OS cross-entropy term yields Equation 8:

$$\mathcal{L}(\theta) \quad = \quad \frac{1}{I} \sum_{i=1}^{I} -P(M_i^{\text{OS}}) \log P_\theta(M_i^{\text{OS}}) + \sum_{m \in M_i^{IS}} -P(m) \log P_\theta(m) \tag{8}$$

In cases where $P(M_i^{\text{OS}}) > 0$, perfectly optimizing $\mathcal{L}_{\text{NLL}}(\theta)$ yields predictions that have (incorrectly) redistributed $P(M_i^{\text{OS}})$ to other peaks. This undesirable behaviour can be avoided by minimizing $\mathcal{L}(\theta)$ instead.

### 4.6 Latent Entropy Regularization

Entropy is a useful tool for interpreting the model's latent distributions. $\mathbb{H}_\theta(N)$ quantifies the diversity of fragments that contribute to the spectrum; $\mathbb{H}_\theta(F|n)$ describes variability in molecular formulae (hydrogen counts) for each fragment; $\mathbb{H}_\theta(N|f)$ describes variability in fragment annotations for each predicted formula. For each latent variable $Z \in \{N, F, F|n, N|f\}$, we define a normalized entropy $\hat{\mathbb{H}}(Z) = \mathbb{H}(Z)/\log(|Z|)$. Normalization corrects for differences in support size, facilitating direct comparison of latent entropies across molecules of different sizes. Since entropy is differentiable, normalized latent entropy and prediction error can be jointly optimized using gradient-based methods. Incorporating normalized entropy into the objective function, as demonstrated in Equation 9, effectively imposes entropy regularization on the latent distributions.

$$\mathcal{L}_{\text{REG}}(\theta) = \mathcal{L}(\theta) + \alpha_N \hat{\mathbb{H}}_\theta(N) + \alpha_F \hat{\mathbb{H}}_\theta(F) + \alpha_{F|n} \hat{\mathbb{H}}_\theta(F|n) + \alpha_{N|f} \hat{\mathbb{H}}_\theta(N|f) \tag{9}$$

The tunable hyperparameters $a_Z \in \{\alpha_N, \alpha_F, \alpha_{F|n}, \alpha_{N|f}\}$ control the influence of the entropy regularizers. Since $\mathcal{L}_{\text{REG}}(\theta)$ is minimized, setting $\alpha_Z < 0$ will maximize the corresponding normalized entropy $\hat{\mathbb{H}}_\theta(Z)$, and vice versa. Entropy regularization can be useful when assessing consistency of fragment annotations, as demonstrated in Section 5.4.

## 5 Experiments

### 5.1 Spectrum Prediction (C2MS)

We evaluated C2MS performance on a held-out portion of the NIST20 MS/MS dataset (Appendix A.9), comparing FraGNNet with other binned and structured prediction models from the literature. ICEBERG (Goldman et al., 2024) is a structured C2MS approach that uses neural networks to generate molecule fragments and map them to a predicted spectrum. GrAFF-MS (Murphy et al., 2023) is a structured approach that predicts a distribution over a static library of common chemical formulae. MassFormer (Young et al., 2023) and NEIMS (Wei et al., 2019) are both binned approaches: the former uses a pretrained graph transformer model (Ying et al., 2021) to encode the molecule, while the latter relies on domain-specific chemical fingerprint representations (Rogers & Hahn, 2010). Precursor-Only is a trivial baseline that only predicts a peak centered on the mass of the precursor formula. For more details on the baseline models, refer to Appendix A.14.

Table 1: Spectrum Prediction Performance on the NIST20 MS/MS Dataset. $C_{\text{BIN}}$ is binned cosine similarity (0.01 Da bins), $C_{\text{HUN}}$ is the Hungarian cosine similarity (10 ppm tolerance). As NEIMS, MassFormer, and ICEBERG have binned outputs, they cannot be scored with $\mathbb{C}_{\text{HUN}}$. Means and standard deviations are reported for 5 random seeds, with best scores in bold.

| Model | InChIKey | | Scaffold | |
| --- | --- | --- | --- | --- |
| | $\mathbb{C}_{\text{BIN}} \Uparrow$ | $\mathbb{C}_{\text{HUN}} \Uparrow$ | $\mathbb{C}_{\text{BIN}} \Uparrow$ | $\mathbb{C}_{\text{HUN}} \Uparrow$ |
| FraGNNet-D4 | **0.736 ± 0.002** | **0.709 ± 0.002** | **0.678 ± 0.005** | **0.654 ± 0.006** |
| FraGNNet-D3 | 0.721 ± 0.002 | 0.693 ± 0.002 | 0.659 ± 0.001 | 0.633 ± 0.001 |
| GrAFF-MS | 0.596 ± 0.005 | 0.602 ± 0.004 | 0.520 ± 0.005 | 0.518 ± 0.004 |
| ICEBERG | 0.707 ± 0.001 | - | 0.636 ± 0.002 | - |
| MassFormer | 0.639 ± 0.002 | - | 0.562 ± 0.003 | - |
| NEIMS | 0.635 ± 0.001 | - | 0.546 ± 0.003 | - |
| Precursor-Only | 0.319 ± 0.000 | 0.285 ± 0.000 | 0.313 ± 0.000 | 0.280 ± 0.000 |

Table 2: Compound Retrieval Performance on the NIST20 MS/MS Dataset. Top-$k$ (for $k \in \{1, 3, 5\}$) is top-$k$ accuracy, expressed as a percentage. Means and standard deviations are reported for 5 random seeds, with best scores in bold.

| Model | InChIKey | | | Scaffold | | |
| --- | --- | --- | --- | --- | --- | --- |
| | Top-1 $\Uparrow$ | Top-3 $\Uparrow$ | Top-5 $\Uparrow$ | Top-1 $\Uparrow$ | Top-3 $\Uparrow$ | Top-5 $\Uparrow$ |
| FraGNNet-D4 | **36.0 ± 0.4** | **66.3 ± 0.5** | **78.4 ± 0.5** | **30.3 ± 0.6** | **59.8 ± 0.3** | **73.2 ± 0.4** |
| ICEBERG | **36.0 ± 0.8** | 63.9 ± 0.3 | 75.4 ± 0.3 | 27.9 ± 0.9 | 55.0 ± 0.7 | 68.3 ± 0.8 |
| MassFormer | 24.2 ± 0.5 | 51.7 ± 0.2 | 65.7 ± 0.6 | 20.4 ± 0.4 | 45.2 ± 0.8 | 60.1 ± 0.8 |
| NEIMS | 26.0 ± 0.7 | 52.7 ± 0.8 | 65.1 ± 0.9 | 19.9 ± 0.8 | 42.0 ± 0.9 | 54.6 ± 0.8 |

The results are summarized in Table 1. FraGNNet-D4, a version of our model that uses a $d = 4$ approximation of the fragmentation DAG, clearly outperformed other models in terms of cosine similarity. Increasing fragmentation depth from $d = 3$ (FraGNNet-D3) had a positive impact on performance, as expected.

## 5.2 Compound Retrieval (MS2C)

Each model was also evaluated in a retrieval-based MS2C task. Our setup is similar to previous works (Goldman et al., 2024; Murphy et al., 2023): for each molecule $X_i$ and associated MS/MS spectrum $Y_i$ in the test set ($\approx 4{,}000$ pairs, Appendix A.9) a candidate set $\mathcal{C}_i$ is constructed from $X_i$ and 49 other molecules sampled from PubChem (Kim et al., 2019). Each $C_i \in \mathcal{C}_i - \{X_i\}$ is selected to have high chemical similarity with $X_i$ as measured by Tanimoto similarity between chemical fingerprints (Rogers & Hahn, 2010). The C2MS models are tasked with predicting a set of spectra $\hat{Y}_i$ for molecules $C_i \in \mathcal{C}_i$. The spectra $\hat{Y}_i$ are ranked by their similarity with the real spectrum $Y_i$, inducing a ranking on the molecules $C_i$. The models are scored based on their ability to correctly rank $X_i$ in the Top-$k$.

For each model, a variety of ranking methods were evaluated, and the one that produced the best ranking was recorded in Table 2. FraGNNet-D4 matched or outperformed all baseline models for all values of $k$ on both splits, with larger gains on the more challenging Scaffold split. For additional retrieval results, refer to Appendix A.11.

## 5.3 Formula Annotations

Peak annotations (Section 2) are an important tool in the interpretation of MS/MS spectra. The FraGNNet model can produce both formula and fragment peak annotations (Figure 3 is an example). In this section we assess the quality of FraGNNet's formula annotations, and in Section 5.4 we evaluate its fragment annotations.

Table 3: Formula Annotations (Scaffold Split). AR is Annotation Recall, AWR is Annotation Weighted Recall, AP is Annotation Precision, AWP is Annotation Weighted Precision, AF is Annotation $F_1$ score, AWF1 is Annotation Weighted $F_1$ score. Means and standard deviations (where applicable) are reported for 5 random seeds, with best scores in bold.

| Model | AR ⇑ | AWR ⇑ | AP ⇑ | AWP ⇑ | AF1 ⇑ | AWF1 ⇑ |
|---|---|---|---|---|---|---|
| FraGNNet-D4 | 0.81 | 0.90 | 0.98 | $\mathbf{1.00 \pm 0.00}$ | **0.89** | $\mathbf{0.95 \pm 0.00}$ |
| GrAFF-MS | **0.95** | **0.98** | 0.61 | $0.81 \pm 0.01$ | 0.74 | $0.88 \pm 0.01$ |
| ICEBERG | $0.66 \pm 0.00$ | $0.81 \pm 0.00$ | $\mathbf{0.99 \pm 0.00}$ | - | $0.79 \pm 0.00$ | - |

The NIST20 dataset provides expert-curated formula annotations for most spectra; for more details on these annotations, refer to Appendix A.15. We pose the formula annotation task as a classification problem. Given a real spectrum $P(m)$, each real peak $m \in M$ has a (possibly empty) set of associated expert formula annotations $A(m) = \{f_m^i\}_i$ that are assumed to be correct. Similarly, in the predicted spectrum $P_\theta(m')$ each predicted peak $m' \in M'$ has a (nonempty) set of associated formula annotations $A(m') = \{f_{m'}^j\}_j$.

Using a relative mass tolerance of 10 ppm, we match peaks in the predicted spectrum $m'$ with peaks in the real spectrum $m$ and identify overlap in their associated formula annotation sets $A(m')$ and $A(m)$. This allows us to calculate recall of annotated peaks in the real spectrum (Equation 31) and precision of annotated peaks in the predicted spectrum (Equation 33), as well as their intensity-weighted counterparts (Equations 32 and 34) and associated $F_1$ scores (Equations 35 and 36).

The results are summarized in Table 3. FraGNNet provided the best balance of recall and precision, as indicated by superior $F_1$ scores. GrAFF-MS achieved slightly higher recall and much lower precision; ICEBERG offered slightly higher precision but lower recall. GrAFF-MS relied on a static formula library, which yielded a distribution over roughly 10,000 formulae for each input compound. By comparison, FraGNNet's formula distribution was typically much smaller, with a median support size of 679 formulae (Table 4). Note that GrAFF-MS directly used NIST formula annotations to define its formula library (refer to Appendix A.14.2 for full details); in contrast, the other models did not rely on expert annotations for training or inference.

## 5.4 Ensembling and Fragment Annotations

Fragment annotations provide greater model interpretability than formula annotations. FraGNNet's latent distribution $P_\theta(n|f)$ naturally provides a map from formulae $f$ to DAG nodes $n$ and their associated fragments $G_n$. $P_\theta(n|f)$ can be interpreted as a fragment annotation distribution, indicating the likelihood of fragment $G_n$ contributing to a peak centered around mass($f$) (see Figure 3 for an example). However, the ground truth distribution $P(n|f)$ is generally unknown and difficult to measure experimentally. In principle, it is possible for multiple fragments (with the same molecular formula) to contribute to the same peak, meaning that the true $P(n|f)$ is not necessarily degenerate.

Unlike the experiments in the previous section, there are no expert-curated fragment annotations that can be used for comparison. Rather than measuring the accuracy of the model annotations, we focus instead on measuring their consistency. In particular, we attempt to construct models with different fragment distributions that nonetheless achieve comparable spectrum prediction performance. Disagreement in fragment annotations between models that perform equally well would indicate that model explanations are variable and therefore potentially unreliable.

We construct four FraGNNet-D4 ensembles, each with a different annotation distribution, by tuning the latent entropy regularization parameters (Section 4.6). The *Baseline Entropy* configuration represents a standard ensemble of $K$ models without entropy regularization ($\alpha_{N|f} = 0$). The *High Entropy* and *Low Entropy* configurations consists of $K$ models with entropy regularization ($\alpha_{N|f} < 0$ and $\alpha_{N|f} > 0$, respectively). The *Mixed Entropy* configuration consists of $K/3$ models each of the Baseline, Low, and High Entropy configurations. All ensembles use $K = 15$ models with the same hyperparameters and different random seeds for initialization and training.

Let $\left\{\theta^k : \theta^k \sim P(\theta)\right\}_{k=1}^K$ denote the set of parameters for an ensemble of $K$ models. We perform ensembling in the latent space (see Appendix A.16 for details). Let $P^K(n|f)$ and $\hat{\mathbb{H}}^K(N|f)$ denote the latent distribution and latent normalized entropy (respectively) of a ensemble of $K$ models. For each configuration, Figure 4a compares the behaviour of the individual models to the overall ensemble, in both the output space and the latent space. Ensembling provides a modest increase in cosine similarity ($4.2 - 4.3\%$) and a somewhat larger increase in normalized entropy ($7.6 - 11.4\%$). The observed entropy increases are consistent with theory; for a short proof see Appendix A.16.

Focusing now on the ensemble metrics, we can see that all four model configurations achieved similar MS/MS spectrum prediction performance ($\mathbb{C}_{\text{HUN}} \approx 0.67$) despite differing markedly in normalized entropy: the Low Entropy configuration only had $\approx 71\%$ of the normalized entropy of the High Entropy configuration ($0.37$ vs $0.52$, respectively). This confirms that FraGNNet models with different fragment annotation distributions can achieve similar C2MS performance.

Viewed in isolation, these results might suggest that FraGNNet's fragment annotations are inconsistent and raise questions about their reliability. However, we also investigated the top-1 fragment agreement (*i.e.*, $P^K(n|f)$ mode consistency) between ensemble configurations. This behaviour was measured using pairwise fragment annotation agreement (PFA, Equation 42) and consensus fragment annotation agreement (CFA, Equation 43). In the NIST20 MS/MS test set, PFA $\geq 82\%$ (Figure 4b) and CFA $\approx 76\%$ across all model configurations. These results demonstrate that the modes of the annotation distributions $P_\theta(n|f)$ were relatively consistent and robust to variations in latent entropy. In Appendix A.17 we performed a similar analysis focusing on $P_\theta(\tilde{n}|f)$, the version of the annotation distribution that accounts for fragment isomorphism (Appendix A.3). In this case the top-1 agreement was even higher (PFA $\geq 91\%$, CFA $\approx 89\%$).

In summary, we have shown that it is possible to achieve comparable performance with ensembles that differ significantly in their normalized latent entropies $\hat{\mathbb{H}}_\theta^K(N|f)$. However, despite these entropy differences, the models tend to agree on which fragment annotation $G_n$ is most appropriate for a given formula $f$ and its associated peak at mass($f$). Although we cannot establish the true fragment annotations without specialized measurements (van Tetering et al., 2024), our experiments indicate that, at a minimum, FraGNNet's explanations remain robust to perturbations in latent entropy.

## 6 Discussion

In this work we introduce FraGNNet, a deep probabilistic model for spectrum prediction. Our work shows that pairing combinatorial fragmentation with graph neural networks can achieve state-of-the-art C2MS performance. FraGNNet is unique in its interpretable probabilistic representation of fragmentation. Features such as OS prediction and tunable entropy regularization further differentiate it from existing models. Strong results in compound retrieval and peak annotation demonstrate potential utility in MS2C applications.

Several paths for improving our method are available. First, the fragmentation algorithm currently depends on recursive edge-removal operations, which could be parallelized to accelerate runtime. Second, the algorithm fails to account for some fragments, as evidenced by unexplained OS peaks in the spectra. These peaks may arise from complex chemistry, such as cyclizations, that dramatically expand the fragment search space and are therefore challenging to model. One potential remedy is to develop a sequential reaction sampler that can generate fragments beyond simple bond-breaking. Alternatively, combining FraGNNet with a more flexible C2MS model, such as a formula predictor like GrAFF-MS, could be an effective method of capturing OS peaks without explicitly modelling their fragments. Finally, to broaden FraGNNet's practical applicability, it is important to evaluate performance on unmerged spectra and under a broader range of experimental conditions, including different instrument platforms and precursor adducts.

## 7 Author Contributions

The project conception was a joint effort of all authors. Experimental design was undertaken by A.Y., F.W., and H.R., with implementation by A.Y. and F.W. All authors participated in revising the manuscript.

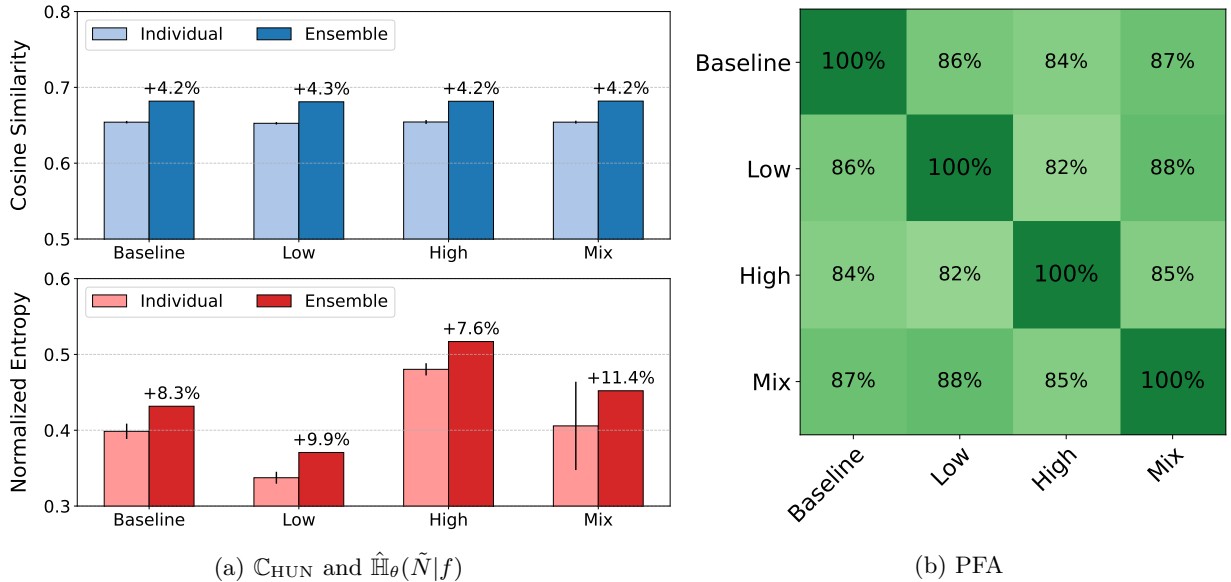

(a) $\mathbb{C}_{\text{HUN}}$ and $\hat{\mathbb{H}}_\theta(\tilde{N}|f)$

(b) PFA

Figure 4: (a) For each ensemble configuration (Baseline, Low, High, Mix), the cosine similarity $\mathbb{C}_{\text{HUN}}$ and normalized entropy of the annotation distribution $\hat{\mathbb{H}}_\theta(N|f)$ are reported. For both metrics, the average score of the individual models (Individual) is compared with the score of the ensemble (Ensemble). Each ensemble consists of $K = 15$ models. Standard deviations for the $K$ individual models are plotted as error bars. (b) Pairwise fragment annotation agreement (PFA) for all ensemble combinations are plotted in a matrix. The consensus agreement (CFA) is $\approx 76\%$.

## 8 Acknowledgements

We thank Tytus Mak for clarification on the techniques used at NIST for formula annotations.

Resources used in preparing this research were provided, in part, by the Province of Ontario, the Government of Canada, through the Canadian Institute for Advanced Research (CIFAR) and companies sponsoring the Vector Institute. This research was also enabled in part by support provided by Compute Ontario (https://www.computeontario.ca/) and the Digital Research Alliance of Canada (alliancecan.ca). A.Y. is supported by a Natural Sciences and Engineering Research Council of Canada (NSERC) Postgraduate Scholarship (Doctoral Program) and a Vector Institute research grant. F.W. is supported by Alberta Machine Intelligence Institute (Amii) research grant. H.R. is supported by NSERC, the Canadian Institutes for Health Research (CIHR), the Canadian Foundation for Innovation, the Canada Research Coordinating Committee (CRCC), the John R. Evans Leaders Fund and the Canada Research Chair Program. B.W. is supported by NSERC (grants: RGPIN-2020-06189 and DGECR-2020-00294), the Peter Munk Cardiac Centre AI Fund at the University Health Network and the CIFAR AI Chair Program. D.S.W. is supported by Genome Canada, Genome British Columbia, and Genome Alberta (project 284MBO); NSERC Brockhouse Canada Prize (NSERC BCPIR-590317) and the Canada Research Chairs (CRC TIER1 100628). R.G. is supported by NSERC Amii and CIFAR AI Chair Program.

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

# A  Appendix

## A.1  Recursive Fragmentation Algorithm

---

**Algorithm 1** RecFrag

---

**Input:** graph $G = (V, E)$, max depth $d$, current depth $d'$
Initialize DAG nodes $V_F = \{\}$, DAG edges $E_F = \{\}$
**if** $d' \leq d$ **then**
    $S = \emptyset$
    $I = \text{ID}(V, E)$
    **for** $e = (u, v) \in E$ **do**
        $E' = E - \{e\}$
        $(V^u, E^u) = \text{BFSCC}(u, V, E')$
        $I^u = \text{ID}(V^u, E^u)$
        $(V^v, E^v) = \text{BFSCC}(v, V, E')$
        $I^v = \text{ID}(V^v, E^v)$
        $S = S \cup \{(V^u, E^u), (V^v, E^v)\}$
        $V_F = V_F \cup \{I^u, I^v\}$
        $E_F = E_F \cup \{(I, I^u), (I, I^v)\}$
    **end for**
    **for** $G^s = (V^s, E^s) \in S$ **do**
        $(V_F^s, E_F^s) = \text{RecFrag}(G^s, d, d' + 1)$
        $V_F = V_F \cup V_F^s$
        $E_F = E_F \cup E_F^s$
    **end for**
**end if**
Return $(V_F, E_F)$

---

The approximate fragmentation DAG $G_{\mathcal{F}^d}$ is constructed by calling Algorithm 1 on the heavy-atom skeleton of the molecule $\mathcal{H}(G)$, with initial depth parameter $d' = 1$. BFSCC is a breadth-first search algorithm that returns the set of nodes and edges in a graph that can be reached from a given input node. We apply BFSCC to identify connected components (fragment subgraphs) of $\mathcal{H}(G)$ after each edge removal. ID is a function that maps every subgraph of $\mathcal{H}(G)$ to a unique integer id (our implementation just uses an enumeration).

We apply a post-processing step that merges fragments $n_1, n_2 \in V_F$ with the same atom sets $V^{n_1} = V^{n_2}$. These fragments arise when bond are removed from non-linear structures in the molecule, such as rings. The merging works as follows: the nodes $\{n_i\}_i$ in the DAG whose associated fragment subgraphs $\{G^{n_i}\}_i$ all have the same atom set $V^n$ are removed from $G_{\mathcal{F}^d}$ and replaced with a new DAG node $m$ that retains all edges of the removed nodes. The associated fragment subgraph $G^m = (V^m, E^m)$ is defined as the vertex-induced subgraph of $V^n$ (Equations 10 and 11):

$$V^m = V^n \tag{10}$$

$$E^m = \{(u, v) \in E : u \in V^n \wedge v \in V^n\} \tag{11}$$

After post-processing, it is possible to assign each node $n$ in the DAG a unique ID based only on its atom set $V^n$. Note that the DAG node merging can introduce self-edges in $G_{\mathcal{F}^d}$, implying that $G_{\mathcal{F}^d}$ is not always a true DAG in practice.

Figure 5 is a visualization of the full fragmentation DAG $\mathcal{F}_G$ for an example molecule. Since this molecule is small and linear, the resulting DAG consists of only 10 nodes and 19 edges.

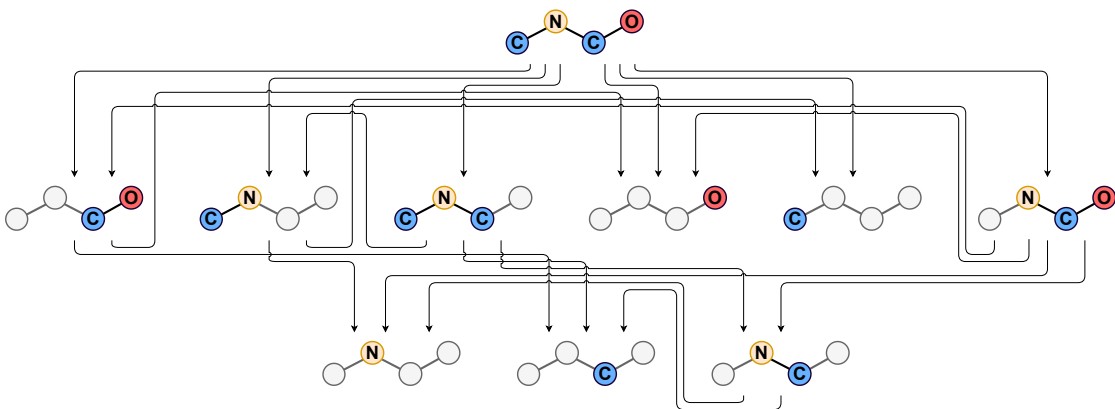

Figure 5: DAG Visualization: the approximate fragmentation DAG $\mathcal{G}_{\mathcal{F}^d}$ ($d = 3$) for a small molecule, methylaminomethanol. The root node is represented by the heavy-atom skeleton of the molecule $\mathcal{H}(G)$, and each other node is represented by a connected subgraph of $\mathcal{H}(G)$. Carbon atoms are represented by C, nitrogen atoms by N, oxygen atoms by O.

Table 4: Fragmentation DAG Statistics over the NIST20 MS/MS Dataset. PP is peak precision, PWP is peak weighted precision, PR is peak recall, PWR is peak weighted recall.

| STATISTICS | D3 ($d=3, j=4$) | | | | | D4 ($d=4, j=4$) | | | | |
|---|---|---|---|---|---|---|---|---|---|---|
| | MIN | 25% | 50% | 75% | MAX | MIN | 25% | 50% | 75% | MAX |
| # FORMULAE $f$ | 4 | 323 | 527 | 838 | 7239 | 4 | 388 | 679 | 1160 | 13146 |
| # NODES $n$ | 10 | 167 | 280 | 474 | 5022 | 10 | 333 | 677 | 1392 | 32902 |
| # NODES $\tilde{n}$ | 9 | 104 | 173 | 308 | 4518 | 9 | 183 | 379 | 865 | 31688 |
| # EDGES $e$ | 20 | 662 | 1156 | 2138 | 44952 | 20 | 2288 | 4790 | 10020 | 249138 |
| PR | 0.00 | 0.51 | 0.64 | 0.77 | 1.00 | 0.00 | 0.63 | 0.75 | 0.85 | 1.00 |
| PWR | 0.00 | 0.75 | 0.88 | 0.96 | 1.00 | 0.00 | 0.86 | 0.94 | 0.98 | 1.00 |
| PP | 0.00 | 0.05 | 0.08 | 0.13 | 0.53 | 0.00 | 0.04 | 0.08 | 0.13 | 0.52 |
| PWP | 0.00 | 0.04 | 0.07 | 0.12 | 0.60 | 0.00 | 0.03 | 0.07 | 0.13 | 0.67 |

## A.2 Fragmentation DAG Statistics

Table 4 describes various distributions related to the size of the approximate DAG $G_{\mathcal{F}^d}$ and its associated mass set $\hat{M}(G, d, j)$ under two different parameterizations. $d = 3, j = 4$ is the configuration for FraGNNet-D3, and $d = 4, j = 4$ is the configuration for FraGNNet-D4.

Let $M$ be the set of masses in the spectrum (peak locations), and let $M' = \hat{M}(G, d, j)$ denote the set of masses given by the approximate DAG for molecule $G$. Peak Recall (PR, Equation 12) is defined as the fraction of peaks in the spectrum that can be explained by a DAG mass while Peak Weighted Recall (PWR, Equation 13) incorporates peak intensities; Peak Precision (PP, Equation 14) and Peak Weighted Precision (PWP, Equation 15) are defined similarly. $\mathbb{I}$ is the indicator function.

$$\mathrm{PR} = \frac{1}{|M|} \sum_{m \in M} \mathbb{I}\left[\exists m \in M' : |m' - m| \leq \epsilon \max(m, 200)\right] \tag{12}$$

$$\mathrm{PWR} = \sum_{m \in M} P(m)\, \mathbb{I}\left[\exists m' \in M' : |m' - m| \leq \epsilon \max(m, 200)\right] \tag{13}$$

$$\text{PP} = \frac{1}{|M'|} \sum_{m' \in M'} \mathbb{I}\left[\exists m \in M : |m' - m| \leq \epsilon \max(m, 200)\right] \tag{14}$$

$$\text{PWP} = \sum_{m' \in M'} P(m') \, \mathbb{I}\left[\exists m \in M : |m' - m| \leq \epsilon \max(m, 200)\right] \tag{15}$$

In our experiments we use a 10 ppm mass tolerance for peak matching ($\epsilon = 10^{-5}$), truncated to a minimum of 0.002 Da.

### A.3 Fragment Subgraph Isomorphism

In MS/MS spectrometry, it is possible for fragments with identical molecular structure to originate from different parts of the molecular graph, having been created through distinct sequences of fragmentation steps. In our model, this phenomenon is represented by pairs of DAG nodes $n_1, n_2 \in V_{\mathcal{F}^d}$ whose corresponding subgraphs $G_{n_1} \cong G_{n_2}$ are isomorphic (*i.e.*, there exists a node bijection between $G_{n_1}$ and $G_{n_2}$ that is both label-preserving and edge-preserving).

With the exception of $P_\theta(f)$ (which does not involve fragments), each of the latent distributions from Section 4.3 can be adapted to account for fragment graph isomorphism. To describe this process precisely, we rely on Definition A.1 and Corollary A.2:

**Definition A.1.** Let $\mathcal{G}$ be a finite set of labelled graphs $\{G_i\}_{i=1}^I$. Each $G_i$ is a member of one of $K$ isomorphism classes $\{\mathcal{G}_k\}_{k=1}^K$, where $K \leq I$. Assume an arbitrary total ordering $\prec_k$ for each isomorphic class $\mathcal{G}_k$. Let $\mathcal{I}(\mathcal{G}) \subseteq \mathcal{G}$ be the set of graphs such that $\forall G_k \in \mathcal{I}(\mathcal{G}) : \nexists G_i \in \mathcal{G} : G_i \in \mathcal{G}_k \wedge G_i \prec_k G_k$.

**Corollary A.2.** $\forall G_i, G_j \in \mathcal{I}(\mathcal{G}) : G_i \neq G_j \rightarrow G_i \not\cong G_j$.

For each DAG node $\tilde{n}$ such that $G_{\tilde{n}} \in \mathcal{I}(\{G_n : n \in V_{\mathcal{F}^d}\})$, we define $P_\theta(\tilde{n})$ as the total probability of all subgraphs isomorphic to $G_{\tilde{n}}$ using Equation 16:

$$P_\theta(\tilde{n}) = \sum_{n \in V_{\mathcal{F}^d} : G_n \cong G_{\tilde{n}}} P_\theta(n) \tag{16}$$

The conditional distributions $P_\theta(f|\tilde{n})$ and $P_\theta(\tilde{n}|f)$ are defined in a similar manner using Equations 17 and 18 respectively:

$$P_\theta(f|\tilde{n}) = \sum_{n \in V_{\mathcal{F}^d} : G_n \cong G_{\tilde{n}}} \frac{P_\theta(f|n) P_\theta(n)}{P_\theta(\tilde{n})} \tag{17}$$

$$P_\theta(\tilde{n}|f) = \sum_{n \in V_{\mathcal{F}^d} : G_n \cong G_{\tilde{n}}} P_\theta(n|f) \tag{18}$$

These distributions can provide additional interpretability, as demonstrated in Appendix A.17. Intuitively, they remove excess entropy caused by uncertainty over the location in the molecule from which each fragment originated.

In practice, we calculate the set $\mathcal{I}(\{G_n : n \in V_{\mathcal{F}^d}\})$ by applying an approximate Weisfeiler-Lehman hashing algorithm (Shervashidze et al., 2011; Hagberg et al., 2008) to each subgraph $G_n : n \in V_{\mathcal{F}^d}$ and identify isomorphism class membership with hash collisions.

### A.4 Molecule Features

Our method for atom and bond featurization follows the approach taken by (Goldman et al., 2024). The features are summarized in Table 5. All discrete features are encoded using a standard one-hot representation. The only continuous feature (Atom Mass) is scaled by a factor of 0.01.

Table 5: Input Features for the Molecule GNN

| Feature | Values |
|---|---|
| Atom Type (Element) | $\{C, O, N, P, S, F, Cl, Br, I, Se, Si\}$ |
| Atom Degree | $\{0, \ldots, 10\}$ |
| Atom Orbital Hybridization | $\{SP, SP2, SP3, SP3D, SP3D2\}$ |
| Atom Formal Charge | $\{-2, \ldots, +2\}$ |
| Atom Radical State | $\{0, \ldots, 4\}$ |
| Atom Ring Membership | $\{True, False\}$ |
| Atom Aromatic | $\{True, False\}$ |
| Atom Mass | $\mathbb{R}^+$ |
| Atom Chirality | $\{Unspecified, Tetrahedral\ CW, Tetrahedral\ CCW\}$ |
| Bond Degree | $\{Single, Double, Triple, Aromatic\}$ |

## A.5 Fragment Features

### A.5.1 Fourier Embeddings

We use Fourier embeddings (Goldman et al., 2023a; Tancik et al., 2020) to represent certain ordinal features such as molecular formulae and collision energy. Given an integer feature $z \in \mathbb{R}$, the corresponding Fourier embedding $\phi(z)$ can be calculated using Equation 19:

$$\phi(z) = \left[ \left| \sin\left( \frac{2\pi z}{\tau_1} \right) \right|, \ldots, \left| \sin\left( \frac{2\pi z}{\tau_T} \right) \right| \right] \tag{19}$$

Compared to a standard one-hot encoding scheme, this approach makes it easier for the model to handle inputs $z$ at inference that have not been seen in training. The periods $\tau_t$ are increasing powers of 2 (we use $T = 10$ for our experiments).

### A.5.2 DAG Node Features

The DAG node embeddings (Equation 4) are initialized with formula information $\hat{h}_n^f$ and fragmentation depth information $\hat{h}_n^d$. The formula embedding for DAG node $n$ is a concatenation of Fourier embeddings corresponding to heavy atom counts in $G_n$, described in Equation 20:

$$\hat{h}_n^f = \bigg\|_{\omega \in \Omega} \phi \left( \sum_{v \in V_n} \mathbb{I}\left[ \omega_v = \omega \right] \right) \tag{20}$$

The depth embedding $\hat{h}_n^d$ is a multi-hot representation of the fragment node's depth in the DAG. The depth set is defined as the set of path lengths between the root node and the fragment node $n$, and is always a subset of $\{0, ..., d\}$ where $d$ is the fragmentation depth.

### A.5.3 DAG Edge Features

The embeddings $\bar{h}_e^{(0)}$ are initialized for each directed edge $e \in E_{\mathcal{F}^d}$ using Equation 21. Assuming $e$ travels from node $n$ to node $t$, let $V_e = V_n - V_t$ be the set of atoms in $G_n$ that are not in $G_t$. As described in Equation 22, $\hat{h}_e^s$ is simply the average of the atom embeddings in $V_e$.

$$\bar{h}_e^{(0)} = \hat{h}_e^s \parallel \hat{h}_e^f \tag{21}$$

$$\hat{h}_e^s = \frac{1}{|V_e|} \sum_{a \in V_e} \bar{h}_a^{(L_1)} \tag{22}$$

The $\hat{h}_e^f$ term is an embedding of the difference of node formulas $f_n - f_t$ using Fourier embeddings, similar to $\hat{h}_n^f$. These features explicitly capture neutral loss information that would otherwise need to be inferred from the DAG.

As previously noted (Section 4.4.2), for most experiments we used a version of the model that omitted DAG edges; in those cases the edge features were not included.

### A.5.4 Collision Energy Embedding

Let $Z_i$ be the set of collision energies that were merged to create spectrum $Y_i$ for molecule $X_i$ in the dataset (Appendix A.10). The collision energy embedding is a representation of $Z_i$. Each collision energy $z \in Z_i$ is a positive integer ranging from 0 to 200 (they are normalized relative to the mass of the precursor, see Young et al. 2023 for more details). The collision energy embedding $\hat{h}_Z$ is simply an average of Fourier embeddings for each collision energy, described by Equation 23:

$$\hat{h}_Z = \frac{1}{|Z|} \sum_{z \in Z} f(z) \tag{23}$$

The collision energy embedding $\hat{h}_Z$ is concatenated with the output Fragment GNN embedding $\bar{h}_n^{(L_2)}$ for each DAG node $n \in G_{\mathcal{F}^d}$ before being passed to the output MLP.

### A.6 Spectrum Similarity Metrics

Mass spectra are typically compared using a form of cosine similarity (Stein & Scott, 1994). The binned approach (Wei et al., 2019; Young et al., 2023; Zhu et al., 2020) involves preprocessing the spectrum by discretizing the mass range into $B$ equally sized bins and summing the intensities for all peaks falling in the same bin. This produces a $B$-dimensional vector of non-negative values that is amenable to standard cosine similarity calculation. In our experiments, binned cosine similarity $\mathbb{C}_{\text{BIN}}$ is calculated with a bin size of 0.01 Da, resulting in $B = 150,000$ (we assume a maximum mass of 1500 Da).

An alternate approach to calculating cosine similarity involves comparing intensities of peaks that are close in mass (Huber et al., 2020; Murphy et al., 2023). This method can be formalized as the linear sum assignment problem below, where $i$ indexes spectrum $Y$, $j$ indexes spectrum $\hat{Y}$ and $p_i, \hat{p}_j$ are shorthand for $P(m_i), \hat{P}(\hat{m}_j)$ respectively:

$$\mathbb{C}_{\text{HUN}}(Y, \hat{Y}) = \max_{w_{ij} \in \{0,1\}} \quad \sum_{i,j} w_{ij} \frac{p_i}{\|p\|_2} \frac{\hat{p}_j}{\|\hat{p}\|_2}$$
$$\text{s.t.} \quad \begin{cases} \sum_i w_{ij} \leq 1 \\ \sum_j w_{ij} \leq 1 \\ |m_i - \hat{m}_j| \leq \tau_i \end{cases} \tag{24}$$

We set the tolerance parameter $\tau_i = 10^{-5} \max(m_i, 200)$ to reflect a thresholded 10 ppm mass error. This maximization problem can be solved efficiently using the Hungarian algorithm (Kuhn, 1955). Note that the tolerance parameter only depends on the masses in spectrum $Y$, introducing an asymmetry in the measure. When comparing a true spectrum with a predicted spectrum (as is typically the case), we use the convention of setting $Y$ to be the true spectrum and $\hat{Y}$ to be the predicted spectrum.

Another important similarity metric is Jensen-Shannon similarity $\mathbb{J}$. Jensen-Shannon similarity is usually defined in terms of the Jensen-Shannon divergence $\mathbb{J}^c$ (Lin, 1991), using Equations 25 and 26:

$$\mathbb{J}(Y, \hat{Y}) = \log 2 - \mathbb{J}^c(Y, \hat{Y}) \tag{25}$$

$$\mathbb{J}^c(Y, \hat{Y}) = H(\bar{Y}) - \frac{1}{2}\left(\mathbb{H}(Y) + \mathbb{H}(\hat{Y})\right) \tag{26}$$

$\mathbb{H}(X)$ is Shannon entropy, log is the natural logarithm, and $\bar{Y}$ is a mixture distribution $\bar{p}_k = 0.5 p_k + 0.5 \hat{p}_k$.

Note that $\mathbb{J}$ can be related to unweighted spectral entropy similarity $\mathbb{S}$ (Li et al., 2021), a common metric in the mass spectrometry literature, through Equation 27.

$$\mathbb{S}(Y, \hat{Y}) = \frac{\mathbb{J}(Y, \hat{Y})}{\log 2} \tag{27}$$

*Proof.* Entropy similarity is defined using Equation 28:

$$\mathbb{S}(Y, \hat{Y}) = 1 - \frac{2H(\bar{Y}) - \mathbb{H}(Y) - \mathbb{H}(\hat{Y})}{\log 4} \tag{28}$$

Equation 27 follows through simple substitution and rearrangement:

$$
\begin{aligned}
\mathbb{S}(Y, \hat{Y}) &= 1 - \frac{2\mathbb{H}(\bar{Y}) - \mathbb{H}(Y) - \mathbb{H}(\hat{Y})}{\log 4} \\
&= 1 - \frac{2\mathbb{J}^c(Y, \hat{Y})}{\log 4} \\
&= 1 - \frac{2(\log 2 - \mathbb{J}(Y, \hat{Y}))}{2\log 2} \\
&= \frac{\mathbb{J}(Y, \hat{Y})}{\log 2}
\end{aligned}
$$

$\square$

### A.7 Out-of-Support Prediction

For a given spectrum, the set of OS peak masses $M^{\mathrm{OS}}$ is defined using a 10 ppm mass cutoff (see Section 4.3). $\delta_{TV}$ is the total variation distance between the true OS distribution and the predicted OS distribution, given by Equation 29:

Table 6: Out-of-Support (OS) Prediction performance on the NIST20 Dataset. Means and standard deviations are reported for 5 random seeds. Best scores (where applicable) are in bold.

| Split | Model | $P(M^{\mathrm{OS}})$ | $P_\theta(M^{\mathrm{OS}})$ | $\delta_{TV}(\downarrow)$ |
|---|---|---|---|---|
| InChIKey | FraGNNet-D3 | $0.171 \pm 0.000$ | $0.191 \pm 0.007$ | $0.080 \pm 0.002$ |
| InChIKey | FraGNNet-D4 | $0.098 \pm 0.000$ | $0.115 \pm 0.003$ | $\mathbf{0.057 \pm 0.001}$ |
| Scaffold | FraGNNet-D3 | $0.193 \pm 0.000$ | $0.213 \pm 0.008$ | $0.094 \pm 0.002$ |
| Scaffold | FraGNNet-D4 | $0.110 \pm 0.000$ | $0.127 \pm 0.002$ | $\mathbf{0.067 \pm 0.001}$ |

$$\delta_{TV} = |P(M^{\mathrm{OS}}) - P_\theta(M^{\mathrm{OS}})| \tag{29}$$

$\delta_{TV}$ measures how well the model can correctly predict the total OS probability. Table 6 summarizes the OS prediction performance of the FraGNNet models. Increasing fragmentation depth from $d = 3$ to $d = 4$ resulted in both a reduction of OS peaks (lower $P(M^{\mathrm{OS}})$) and an improved ability to approximate $P(M^{\mathrm{OS}})$ with $P_\theta(M^{\mathrm{OS}})$. This is useful since knowing $P(M^{\mathrm{OS}})$ at inference time can help identify situations where the fragmentation algorithm performs poorly (in terms of PR and PWR, Section A.2). In such cases, it is impossible for the model to make an accurate MS/MS prediction.

## A.8 Model Ablations

We performed two kinds of ablations on FraGNNet: (-CE) corresponds to the removal of merged collision energy information (Appendix A.5.4), (+Edges) corresponds to the addition of bidirectional edges (and associated edge embeddings) and message passing in the Fragment GNN. The removal of collision energy information had a negative impact on performance, as expected. However, the addition of DAG edge information did not have a strong effect. This seems to suggest either that the hierarchical relationships between fragments (*i.e.*, nodes in the DAG) might be easy for the model to infer without DAG edges, or that this information is not helpful for making spectrum predictions.

ICEBERG (+OptFrag) is a modified version of ICEBERG that uses additional information from the ground-truth MS/MS spectrum to optimally select a set of fragments for each prediction. We emphasize that this ablation is intended solely for benchmarking purposes, since in most real-world applications the goal is to make predictions for molecules without first observing their MS/MS spectra. Providing ICEBERG with this additional information resulted in improved performance that nearly matched that of FraGNNet-D4 (Table 8), confirming that autoregressive fragment generation is a source of error for ICEBERG.

## A.9 Datasets and Splits

We trained and evaluated all models on the NIST 2020 MS/MS dataset (Stein, 2012; Yang et al., 2014), a large commercial library of MS data. To ensure homogeneity, the original dataset was filtered to include only $[\mathrm{M+H}]^+$ adducts. Following (Goldman et al., 2023a; 2024), spectra for the same compound acquired at different collision energies were combined (a process commonly referred to as *collision energy merging*, see Appendix A.10). The resulting dataset contained 21,113 unique molecules, each with an associated merged MS/MS spectrum. Models were trained using 60% of the data, with 20% for validation and 20% used as a heldout test set. Two strategies for data splitting were employed: a simple random split by molecule ID using the InChIKey hashing algorithm (Heller et al., 2015), and a more challenging split that clustered molecules based on their Murcko Scaffold (a coarse representation of 2D molecular structure, Bemis & Murcko 1996). Scaffold splits are commonly used to evaluate generalization of deep learning models in cheminformatics applications (Wu et al., 2018).

Table 7: FraGNNet Model Ablations. (-CE) is removal of collision energy covariates, (+Edges) is the addition of DAG edges and associated features. Performance reported on the InChIKey test set (mean and standard deviation of 5 random seeds). Best scores are in bold.

| MODEL | $\mathbb{C}_{\mathrm{BIN}} \Uparrow$ | $\mathbb{C}_{\mathrm{HUN}} \Uparrow$ |
|---|---|---|
| FRAGNNET-D3 | $\mathbf{0.721 \pm 0.002}$ | $\mathbf{0.693 \pm 0.002}$ |
| FRAGNNET-D3 (-CE) | $0.711 \pm 0.001$ | $0.683 \pm 0.001$ |
| FRAGNNET-D3 (+EDGES) | $0.716 \pm 0.001$ | $0.687 \pm 0.001$ |
| FRAGNNET-D4 | $\mathbf{0.736 \pm 0.002}$ | $\mathbf{0.709 \pm 0.002}$ |
| FRAGNNET-D4 (-CE) | $0.721 \pm 0.001$ | $0.694 \pm 0.001$ |

Table 8: ICEBERG Model Ablations. (+OptFrag) replaces learned fragment generation with an optimal approach that uses information from the ground-truth MS/MS spectrum. Performance reported on both InChIKey and Scaffold test sets (mean and standard deviation of 5 random seeds). Best scores are in bold.

| MODEL | INCHIKEY $\mathbb{C}_{\text{BIN}} \Uparrow$ | SCAFFOLD $\mathbb{C}_{\text{BIN}} \Uparrow$ |
|---|---|---|
| FRAGNNET-D4 | **0.736 ± 0.002** | **0.678 ± 0.005** |
| ICEBERG | 0.707 ± 0.001 | 0.636 ± 0.002 |
| ICEBERG (+OPTFRAG) | **0.732 ± 0.000** | **0.668 ± 0.002** |

### A.10 Data Preprocessing

We exported spectra from the NIST20 MS/MS spectral library (following this github repository). We applied a number of filters based on the spectral metadata and molecular properties.

The metadata criteria are described below:

- Orbitrap instrument with higher-energy collisional dissociation (HCD)

- $[M+H]^+$ adduct type

- Precursor m/z $\leq 1500$

- Normalized collision energy

The molecule criteria are described below:

- Element composition: $H, C, O, N, P, S, F, Cl, Br, I, Se, Si$

- $\leq 60$ heavy (non-H) atoms

- Neutral charge

- No radical electrons

- Single molecule

After filtering, there were 262,319 spectra representing 21,113 molecules.

Individual MS/MS spectra for the same molecule and precursor adduct were merged across collision energies by averaging the mass distributions. More formally, let $\{Y^{(i)}\}_{i=1}^I$ be the set of spectra corresponding to molecule $X$ (each measured with a different collision energy $Z_i$). The merged spectrum $Y$ is defined using Equation 30.

$$Y = \bigcup_{i=1}^{I} \left\{ \left( m_j^{(i)}, \frac{P(m_j)^{(i)}}{I} \right) \right\}_j^{(i)} \tag{30}$$

### A.11 Compound Retrieval Continued

We investigated the effect of different *ranking methods* on retrieval performance (Section 5.2). A ranking method is defined as a combination of a spectrum similarity metric, a peak intensity transformation, and a peak matching strategy. For similarity metrics, the choices are cosine similarity $\mathbb{C}$ or Jensen-Shannon similarity $\mathbb{J}$. For peak intensity transformations, the choices are square-root or identity function (no transformation).

Table 9: Compound Retrieval Results Extended (InChIKey Split). Top-$k$ (for $k \in \{1, 2, 3, 5, 10\}$) is top-$k$ accuracy, expressed as a percentage; MRR is mean reciprocal rank. Means and standard deviations are reported for 5 random seeds; results reported in Table 2 are in bold.

| Model | Method | Top-1 ⇑ | Top-2 ⇑ | Top-3 ⇑ | Top-5 ⇑ | Top-10 ⇑ | MRR ⇑ |
|---|---|---|---|---|---|---|---|
| FraGNNet-D4 | $\mathbb{C}_{\text{BIN}}$ | $25.3 \pm 0.5$ | $41.2 \pm 0.5$ | $52.6 \pm 0.9$ | $67.0 \pm 0.6$ | $83.7 \pm 0.4$ | $0.436 \pm 0.003$ |
| FraGNNet-D4 | $\mathbb{C}_{\text{BIN}} + \sqrt{}$ | $35.0 \pm 0.6$ | $53.9 \pm 0.7$ | $65.6 \pm 0.2$ | $78.0 \pm 0.4$ | $90.3 \pm 0.4$ | $0.534 \pm 0.004$ |
| FraGNNet-D4 | $\mathbb{C}_{\text{HUN}}$ | $25.4 \pm 0.4$ | $41.5 \pm 0.2$ | $53.1 \pm 0.7$ | $67.5 \pm 0.5$ | $84.3 \pm 0.4$ | $0.438 \pm 0.002$ |
| FraGNNet-D4 | $\mathbb{C}_{\text{HUN}} + \sqrt{}$ | $35.7 \pm 0.4$ | $54.1 \pm 0.6$ | $65.9 \pm 0.4$ | $78.3 \pm 0.6$ | $90.4 \pm 0.5$ | $0.539 \pm 0.003$ |
| FraGNNet-D4 | $\mathbb{J}_{\text{BIN}}$ | $34.4 \pm 0.4$ | $53.2 \pm 0.7$ | $64.8 \pm 0.3$ | $77.5 \pm 0.4$ | $89.9 \pm 0.4$ | $0.529 \pm 0.003$ |
| FraGNNet-D4 | $\mathbb{J}_{\text{BIN}} + \sqrt{}$ | $36.1 \pm 0.4$ | $54.3 \pm 0.4$ | $65.4 \pm 0.4$ | $77.8 \pm 0.2$ | $89.8 \pm 0.3$ | $0.54 \pm 0.002$ |
| FraGNNet-D4 | $-\mathcal{L}$ | $\mathbf{36.0 \pm 0.4}$ | $54.1 \pm 0.6$ | $\mathbf{66.3 \pm 0.5}$ | $\mathbf{78.4 \pm 0.5}$ | $90.4 \pm 0.2$ | $0.541 \pm 0.004$ |
| ICEBERG | $\mathbb{C}_{\text{BIN}}$ | $27.0 \pm 0.4$ | $43.2 \pm 0.4$ | $53.6 \pm 0.6$ | $66.2 \pm 0.5$ | $81.7 \pm 0.2$ | $0.446 \pm 0.003$ |
| ICEBERG | $\mathbb{C}_{\text{BIN}} + \sqrt{}$ | $34.8 \pm 0.4$ | $52.2 \pm 1.0$ | $63.2 \pm 0.4$ | $74.7 \pm 0.4$ | $87.4 \pm 0.4$ | $0.523 \pm 0.004$ |
| ICEBERG | $\mathbb{J}_{\text{BIN}}$ | $34.4 \pm 0.3$ | $51.5 \pm 0.8$ | $62.3 \pm 0.5$ | $73.9 \pm 0.5$ | $86.8 \pm 0.3$ | $0.517 \pm 0.003$ |
| ICEBERG | $\mathbb{J}_{\text{BIN}} + \sqrt{}$ | $\mathbf{36.0 \pm 0.8}$ | $53.4 \pm 0.6$ | $\mathbf{63.9 \pm 0.3}$ | $\mathbf{75.4 \pm 0.3}$ | $87.2 \pm 0.4$ | $0.532 \pm 0.005$ |
| ICEBERG | $-\mathcal{L}$ | $34.8 \pm 0.4$ | $52.2 \pm 1.0$ | $63.2 \pm 0.4$ | $74.7 \pm 0.4$ | $87.4 \pm 0.4$ | $0.523 \pm 0.004$ |
| MassFormer | $\mathbb{C}_{\text{BIN}}$ | $16.5 \pm 0.7$ | $29.6 \pm 0.3$ | $39.1 \pm 0.5$ | $51.5 \pm 0.5$ | $70.2 \pm 0.5$ | $0.332 \pm 0.003$ |
| MassFormer | $\mathbb{C}_{\text{BIN}} + \sqrt{}$ | $23.7 \pm 0.4$ | $39.3 \pm 0.4$ | $50.4 \pm 0.5$ | $63.9 \pm 0.1$ | $80.5 \pm 0.2$ | $0.417 \pm 0.002$ |
| MassFormer | $\mathbb{J}_{\text{BIN}}$ | $22.9 \pm 0.3$ | $38.3 \pm 0.3$ | $49.4 \pm 0.4$ | $63.0 \pm 0.2$ | $79.9 \pm 0.2$ | $0.408 \pm 0.002$ |
| MassFormer | $\mathbb{J}_{\text{BIN}} + \sqrt{}$ | $\mathbf{24.2 \pm 0.5}$ | $40.7 \pm 0.3$ | $\mathbf{51.7 \pm 0.2}$ | $\mathbf{65.7 \pm 0.6}$ | $82.8 \pm 0.3$ | $0.426 \pm 0.002$ |
| MassFormer | $-\mathcal{L}$ | $23.7 \pm 0.4$ | $39.3 \pm 0.4$ | $50.4 \pm 0.5$ | $63.9 \pm 0.1$ | $80.5 \pm 0.2$ | $0.417 \pm 0.002$ |
| NEIMS | $\mathbb{C}_{\text{BIN}}$ | $18.3 \pm 0.5$ | $30.7 \pm 0.2$ | $39.5 \pm 0.5$ | $51.3 \pm 0.8$ | $68.8 \pm 0.5$ | $0.341 \pm 0.004$ |
| NEIMS | $\mathbb{C}_{\text{BIN}} + \sqrt{}$ | $25.1 \pm 0.6$ | $40.2 \pm 0.8$ | $50.1 \pm 0.8$ | $62.5 \pm 1.0$ | $78.1 \pm 0.6$ | $0.421 \pm 0.006$ |
| NEIMS | $\mathbb{J}_{\text{BIN}}$ | $24.7 \pm 0.5$ | $39.6 \pm 0.8$ | $49.4 \pm 0.8$ | $61.6 \pm 1.0$ | $77.2 \pm 0.6$ | $0.415 \pm 0.006$ |
| NEIMS | $\mathbb{J}_{\text{BIN}} + \sqrt{}$ | $\mathbf{26.0 \pm 0.7}$ | $42.1 \pm 1.1$ | $\mathbf{52.7 \pm 0.8}$ | $\mathbf{65.1 \pm 0.9}$ | $80.1 \pm 0.6$ | $0.436 \pm 0.007$ |
| NEIMS | $-\mathcal{L}$ | $25.1 \pm 0.6$ | $40.2 \pm 0.8$ | $50.1 \pm 0.8$ | $62.5 \pm 1.0$ | $78.1 \pm 0.6$ | $0.421 \pm 0.006$ |

For peak matching, the choices are 0.01 Da binning (BIN) or Hungarian matching with 10ppm tolerance (HUN). Refer to Appendix A.6 for more details on similarity metrics and peak matching strategies. Since FraGNNet was the only model in this experiment that does not output binned $m/z$ values, none of the other baseline models were evaluated with Hungarian peak matching.

In addition to the ranking methods described above, the negative of the loss function (denoted as $-\mathcal{L}$) is also reported. Note that loss functions are model-dependent and may not be symmetric: in this retrieval evaluation, we use the convention that the query spectrum is ground truth and the reference spectrum is the prediction. This is relevant for FraGNNet-D4, which uses an asymmetric cross-entropy loss function (Equation 8). The other models use binned cosine distance loss with square root intensity transform, so the performance of the $-\mathcal{L}$ ranking method is identical to the $\mathbb{C}_{\text{BIN}} + \sqrt{}$ ranking method.

In Tables 9 and 10, each model's retrieval performance is reported for all ranking methods. The configurations reported in Table 2 in the main text are highlighted in bold.

## A.12   Implementation Details

The FraGNNet model and baselines were implemented in Python (Python Core Team, 2021), using Pytorch (Paszke et al. 2019, version 2.1 with CUDA 11.8) and Pytorch Lightning (Falcon & The PyTorch Lightning team, 2019). Weights and Biases (Biewald, 2020) was used to track experiments and run hyperparameter sweeps. The recursive fragmentation algorithm was implemented in Cython (Behnel et al., 2011). The graph neural network modules were implemented using Pytorch Geometric (Fey & Lenssen, 2019). The data preprocessing and molecule featurization used RDKit (Landrum, 2022).

The code is available at github.com/FraGNNet/fragnnet.

Table 10: Compound Retrieval Results Extended (Scaffold Split). Top-$k$ (for $k \in \{1, 2, 3, 5, 10\}$) is top-$k$ accuracy, expressed as a percentage; MRR is mean reciprocal rank. Means and standard deviations are reported for 5 random seeds; results reported in Table 2 are indicated in bold.

| Model | Method | Top-1 ⇑ | Top-2 ⇑ | Top-3 ⇑ | Top-5 ⇑ | Top-10 ⇑ | MRR ⇑ |
|---|---|---|---|---|---|---|---|
| FraGNNet-D4 | $\mathbb{C}_{\text{BIN}}$ | $21.0 \pm 0.7$ | $35.8 \pm 0.8$ | $46.8 \pm 0.4$ | $61.4 \pm 1.0$ | $79.7 \pm 0.5$ | $0.391 \pm 0.006$ |
| FraGNNet-D4 | $\mathbb{C}_{\text{BIN}} + \sqrt{}$ | $29.6 \pm 0.6$ | $47.9 \pm 0.6$ | $59.4 \pm 0.4$ | $73.0 \pm 0.3$ | $87.6 \pm 0.3$ | $0.485 \pm 0.004$ |
| FraGNNet-D4 | $\mathbb{C}_{\text{HUN}}$ | $21.1 \pm 0.7$ | $36.5 \pm 0.9$ | $47.8 \pm 0.5$ | $62.3 \pm 0.7$ | $80.5 \pm 0.4$ | $0.395 \pm 0.006$ |
| FraGNNet-D4 | $\mathbb{C}_{\text{HUN}} + \sqrt{}$ | $\mathbf{30.3 \pm 0.6}$ | $48.2 \pm 0.7$ | $\mathbf{59.8 \pm 0.3}$ | $\mathbf{73.2 \pm 0.4}$ | $87.6 \pm 0.3$ | $0.489 \pm 0.004$ |
| FraGNNet-D4 | $\mathbb{J}_{\text{BIN}}$ | $28.9 \pm 0.9$ | $46.8 \pm 0.8$ | $58.4 \pm 0.3$ | $72.3 \pm 0.3$ | $87.2 \pm 0.2$ | $0.477 \pm 0.006$ |
| FraGNNet-D4 | $\mathbb{J}_{\text{BIN}} + \sqrt{}$ | $29.7 \pm 0.8$ | $47.8 \pm 1.0$ | $59.4 \pm 0.4$ | $73.0 \pm 0.4$ | $86.9 \pm 0.4$ | $0.484 \pm 0.005$ |
| FraGNNet-D4 | $-\mathcal{L}$ | $29.8 \pm 1.0$ | $47.1 \pm 1.0$ | $59.2 \pm 0.6$ | $73.2 \pm 0.7$ | $87.7 \pm 0.5$ | $0.484 \pm 0.008$ |
| ICEBERG | $\mathbb{C}_{\text{BIN}}$ | $20.4 \pm 0.5$ | $33.8 \pm 0.4$ | $44.2 \pm 0.9$ | $58.2 \pm 0.8$ | $76.4 \pm 0.8$ | $0.376 \pm 0.004$ |
| ICEBERG | $\mathbb{C}_{\text{BIN}} + \sqrt{}$ | $27.0 \pm 0.6$ | $42.8 \pm 0.3$ | $54.1 \pm 0.3$ | $68.0 \pm 0.5$ | $83.6 \pm 0.6$ | $0.449 \pm 0.003$ |
| ICEBERG | $\mathbb{J}_{\text{BIN}}$ | $26.4 \pm 0.7$ | $41.9 \pm 0.4$ | $52.9 \pm 0.4$ | $66.6 \pm 0.5$ | $82.9 \pm 0.6$ | $0.442 \pm 0.004$ |
| ICEBERG | $\mathbb{J}_{\text{BIN}} + \sqrt{}$ | $\mathbf{27.9 \pm 0.9}$ | $43.8 \pm 0.4$ | $\mathbf{55.0 \pm 0.7}$ | $\mathbf{68.3 \pm 0.8}$ | $83.6 \pm 0.9$ | $0.456 \pm 0.007$ |
| ICEBERG | $-\mathcal{L}$ | $27.0 \pm 0.6$ | $42.8 \pm 0.3$ | $54.1 \pm 0.3$ | $68.0 \pm 0.5$ | $83.6 \pm 0.6$ | $0.449 \pm 0.003$ |
| MassFormer | $\mathbb{C}_{\text{BIN}}$ | $12.9 \pm 0.3$ | $23.2 \pm 1.1$ | $31.8 \pm 1.1$ | $45.5 \pm 1.3$ | $65.7 \pm 1.3$ | $0.286 \pm 0.006$ |
| MassFormer | $\mathbb{C}_{\text{BIN}} + \sqrt{}$ | $19.1 \pm 0.5$ | $33.1 \pm 0.7$ | $43.9 \pm 0.4$ | $58.2 \pm 0.7$ | $76.9 \pm 1.0$ | $0.369 \pm 0.005$ |
| MassFormer | $\mathbb{J}_{\text{BIN}}$ | $18.6 \pm 0.3$ | $32.2 \pm 0.6$ | $42.6 \pm 1.0$ | $57.2 \pm 0.8$ | $76.0 \pm 0.8$ | $0.361 \pm 0.005$ |
| MassFormer | $\mathbb{J}_{\text{BIN}} + \sqrt{}$ | $\mathbf{20.4 \pm 0.4}$ | $34.6 \pm 0.5$ | $\mathbf{45.2 \pm 0.8}$ | $\mathbf{60.1 \pm 0.8}$ | $78.9 \pm 1.1$ | $0.383 \pm 0.005$ |
| MassFormer | $-\mathcal{L}$ | $19.1 \pm 0.5$ | $33.1 \pm 0.7$ | $43.9 \pm 0.4$ | $58.2 \pm 0.7$ | $76.9 \pm 1.0$ | $0.369 \pm 0.005$ |
| NEIMS | $\mathbb{C}_{\text{BIN}}$ | $12.7 \pm 0.4$ | $22.3 \pm 0.3$ | $30.1 \pm 0.5$ | $41.6 \pm 0.4$ | $59.8 \pm 0.7$ | $0.273 \pm 0.003$ |
| NEIMS | $\mathbb{C}_{\text{BIN}} + \sqrt{}$ | $18.4 \pm 0.9$ | $30.9 \pm 0.6$ | $40.1 \pm 0.5$ | $52.8 \pm 0.4$ | $70.8 \pm 0.3$ | $0.347 \pm 0.007$ |
| NEIMS | $\mathbb{J}_{\text{BIN}}$ | $17.7 \pm 1.0$ | $29.9 \pm 0.7$ | $39.1 \pm 0.5$ | $51.7 \pm 0.6$ | $70.0 \pm 0.2$ | $0.339 \pm 0.007$ |
| NEIMS | $\mathbb{J}_{\text{BIN}} + \sqrt{}$ | $\mathbf{19.9 \pm 0.8}$ | $32.7 \pm 0.8$ | $\mathbf{42.0 \pm 0.9}$ | $\mathbf{54.6 \pm 0.8}$ | $72.8 \pm 0.6$ | $0.363 \pm 0.007$ |
| NEIMS | $-\mathcal{L}$ | $18.4 \pm 0.9$ | $30.9 \pm 0.6$ | $40.1 \pm 0.5$ | $52.8 \pm 0.4$ | $70.8 \pm 0.3$ | $0.347 \pm 0.007$ |

## A.13 Parameter Counts

Table 11: Parameter counts, reported in millions.

| MODEL | # PARAMETERS |
|---|---|
| FRAGNNET-D4 | 1.6 |
| FRAGNNET-D3 | 1.2 |
| ICEBERG | 17.6 |
| MASSFORMER | 165.3 |
| NEIMS | 125.4 |
| GRAFF-MS | 57.8 |

The parameter counts for all models are summarized in Table 11.

## A.14 Baseline Models

All baseline models were re-implemented in our framework, to facilitate fair comparison across methods. Some of the models were originally designed to support additional covariates such as precursor adduct and instrument type. Since our experiments were restricted to MS/MS data of a single precursor adduct ($[\text{M+H}]^+$) and instrument type (Orbitrap), we excluded these features in our implementations.

### A.14.1 ICEBERG

ICEBERG (Inferring Collision-induced-dissociation by Estimating Breakage Events and Reconstructing their Graphs, Goldman et al. 2024) is a state-of-the-art C2MS model. ICEBERG is composed of two sub-modules (neural networks) that are trained independently of each other. The first module (the *fragment generator*, also called ICEBERG-Generate) autoregressively predicts a simplified fragmentation DAG, and the second module (the *intensity predictor*, also called ICEBERG-Score) outputs a distribution over those fragments. The fragment generator is trained to approximate a fragmentation tree that is constructed using a variant of the MAGMa algorithm (Ridder et al., 2014). MAGMa applies a combinatorial atom removal strategy to generate a fragmentation DAG from an input molecular graph $G$: the resulting MAGMa DAG has a similar set of fragment nodes to $G_{\mathcal{F}^d}$, but may contain different edges (refer to Ridder et al. 2014; Goldman et al. 2024 for full details). The MAGMa DAG is then simplified using a number of pruning strategies. Fragments with masses that are not represented by any peak in the spectrum are removed. Fragments that map to same peak are deduplicated: chemical heuristics are used to determine which fragment should be kept. Unlike our approach, no distinction is made between isomorphic fragments that originate from different parts of the molecule. Redundant paths between fragments are removed to convert the DAG into a proper tree, which is required for autoregressive generation.

The aggressive DAG pruning removes information that could be important for correctly predicting the spectrum. However, this pruning also facilitates more expressive representations of the fragments that remain: since the total number of fragments is lower, the computational and memory cost per fragment can be much higher. This tradeoff underlines the key conceptual difference between FraGNNet and ICEBERG. The former uses a more complete fragmentation DAG but must employ a simpler representation for each individual fragment. The latter can afford a more complex fragment representation but only considers a sampled subset of the DAG.

In ICEBERG the output spectrum is binned, with each fragment contributing to the intensity of a particular bin (refer to Goldman et al. 2024, Section 2.5 for full details). Unlike FraGNNet, ICEBERG does not explicitly model a latent fragment distribution $P_\theta(n)$ or formula distribution $P_\theta(f)$. There is no straightforward method to calculate these latent distributions in a way that is consistent with the output spectrum.

ICEBERG (+OptFrag) is a variant of ICEBERG that we introduced for the purposes of benchmarking (Section A.8). ICEBERG (+OptFrag) replaces the learned fragment generator with the exact output of the MAGMa algorithm. This modification removes sampling error in the fragment generation process, creating an artificially easier learning problem that should result in better performance. However, it assumes access to the ground truth MS/MS spectrum for the input molecule. ICEBERG (+OptFrag) is helpful in benchmarking because it can be used to infer how much overall MS/MS prediction error is caused by incorrect fragment generation, but it is not practically useful as a C2MS model.

Our ICEBERG implementation was based on the code from Goldman et al. 2024, although in our experiments the model was trained using binned cosine similarity with 0.01 Da bins (previously, it was trained using 0.1 Da bins). We applied a square root transformation to each binned target intensity during training. We also fixed a bug that caused spurious peaks to appear in the smallest m/z bin as a result of improper batching.

### A.14.2 GrAFF-MS

GrAFF-MS (Graph neural network for Approximation via Fixed Formulas of Mass Spectra, Murphy et al. 2023) is a structured C2MS model. Unlike ICEBERG and FraGNNet, GrAFF-MS does not rely on fragment information, instead predicting a distribution over formulae $P_\theta(f)$ directly. It uses a static library of common product and neutral loss formulae derived from the formula annotations of a labelled spectrum dataset. At inference time, the model predicts a distribution over the formula library, and the formula masses are used to map this distribution to a spectrum.

In our experiments we used the NIST20 expert-curated formula annotations to construct the formula library (see Appendix A.15 for details). Each formula annotation $f$ in the dataset can be interpreted as a product formula $f^+ = f$; the associated neutral loss formula can be calculated as $f^- = p - f$, where $p$ is the precursor formula for that spectrum. For each formula annotation in the dataset, we recorded its associated peak

intensity. In the case of multiple formula annotations for the same peak, we divided the peak intensity equally across all annotations. We then selected the top 10,000 product and neutral loss formulae in terms of total intensity across spectra in the dataset. Given a molecule with precursor formula $p$, the support of the formula distribution $P_\theta(f)$ is defined as the union of the set of product formulae $\{f_i^+\}_i$ and the set of complements of the neutral loss formulae $\{p - f_j^-\}_j$.

We trained the model with a peak-marginal cross-entropy loss (Murphy et al., 2023) using a thresholded 10 ppm mass tolerance (to be consistent with our $C_{HUN}$ metric). The original GrAFF-MS implementation was designed to work with a single collision energy; our implementation uses the average collision energy to predict the merged spectrum. Like the original model, our implementation supports isotopic traces for each predicted peak, modelled as an additional covariate (*i.e.*, binary variable indicating presence/absence of isotopes). This sort of isotope information is present in the NIST20 MS/MS dataset, although it is not used for FraGNNet or the other baseline models.

### A.14.3  NEIMS

NEIMS (Neural Electron Ionization Mass Spectrometry, Wei et al. 2019) was the first deep learning C2MS model, originally designed for low mass accuracy (1.0 Da bins) electron-ionization mass spectrometry (EI-MS) prediction. NEIMS represents the input molecule using a domain-specific featurization method called a molecular fingerprint, which capture useful properties of the molecule such as the presence or absence of various substructures. More recent works (Zhu et al., 2020; Young et al., 2023; Goldman et al., 2023a) have adapted NEIMS to ESI-MS/MS prediction at higher mass accuracy (0.1 Da bins). We based our implementation on the version from Young et al. 2023 which uses three different kinds of molecular fingerprint representations: the Extended Connectivity (Morgan) fingerprint (Rogers & Hahn, 2010), the RDKit fingerprint (Landrum, 2022), and the Molecular Access Systems (MACCS) fingerprint (Durant et al., 2002). We adapted the model to use the same collision energy featurization strategy as FraGNNet (Equation 23). Finally, to avoid an excess of parameters, we replaced the final fully-connected layer's weight matrix with a low-rank approximation. This layer maps from the latent dimension $d_h = 1024$ to the output dimension $d_o = 150000$, corresponding to the mass range $[0, 1500]$ with 0.01 Da bins. The low-rank approximation was implemented as product of two learnable weight matrices: a $d_h \times d_r$ matrix and a $d_r \times d_o$ matrix, where $d_r = 256$.

We trained the model with binned cosine similarity, and applied a square root transformation to the binned target intensities.

### A.14.4  MassFormer

MassFormer (Young et al., 2023) is a binned C2MS method that was originally designed for low mass accuracy (1.0 Da bins) MS/MS prediction. It uses a graph transformer architecture (Ying et al., 2021) that is pre-trained on a large chemical dataset (Nakata & Shimazaki, 2017) and then fine-tuned on spectrum prediction. We preserved most aspects of the model's original implementation but adapted the collision energy featurization and low rank output matrix approximation ($d_r = 128$) from the NEIMS baseline. Unlike the original MassFormer paper, we did not employ FLAG (Kong et al., 2022), a strategy for adversarial data augmentation in graphs, since none of the other models used data augmentation. We trained the model with binned cosine similarity, and applied a square root transformation to the binned target intensities.

### A.14.5  Precursor-Only

A trivial baseline that simply puts 100% of the intensity on the precursor peak. More formally, given the input molecule's precursor formula $p$, the model predicts a spectrum with a single peak $\{(\text{mass}(p), 1)\}$. This baseline is useful because the precursor peak typically accounts for a large fraction of the overall intensity in the spectrum.

### A.15 Formula Annotation Evaluation

The NIST20 expert formula annotations were determined using a combination of algorithmic and manual approaches. Following (Murphy et al., 2023), we excluded glycan and peptide spectra from our formula annotation analysis and focused on small molecules.

Based on our correspondence with scientists at NIST, the procedure for establishing small molecule formula annotations can be described as follows. For each precursor compound, an exhaustive decomposition of the precursor molecular formula was used to enumerate all possible subformulae. These subformulae were then matched to peaks in the spectrum by comparing exact subformula masses with measured peak masses. Finally, expert opinion was used to remove unlikely annotations.

While the details of the matching process are not published, our analysis indicates that the vast majority ($\approx 95\%$) of formula annotation masses were within 10 ppm of their associated real peak masses. A small fraction of annotation formula masses ($< 1\%$) are within 10 ppm of at least one other peak in the spectrum. For consistency with the peak matching metrics used elsewhere in the paper (*i.e.*, the Hungarian cosine similarity metric $\mathbb{C}_{\mathrm{HUN}}$, Appendix A.6), we excluded annotations with formula masses more than 10 ppm away from their associated peak mass.

The annotation metrics used in Section 5.3 are defined in the following equations; $\epsilon = 10^{-5}$ is the mass tolerance parameter.

$$\mathrm{AR} = \frac{\sum_{m \in M} \mathbb{I}\left[\exists m' \in M' : (|m - m'| \leq \epsilon \max(m, 200)) \wedge (|A(m) \wedge A(m')| > 0)\right]}{\sum_{m \in M} \mathbb{I}\left[|A(m)| > 0\right]} \tag{31}$$

$$\mathrm{AWR} = \frac{\sum_{m \in M} P(m)\mathbb{I}\left[\exists m' \in M' : (|m - m'| \leq \epsilon \max(m, 200)) \wedge (|A(m) \wedge A(m')| > 0)\right]}{\sum_{m \in M} P(m)\mathbb{I}\left[|A(m)| > 0\right]} \tag{32}$$

$$\mathrm{AP} = \frac{\sum_{m' \in M'} \mathbb{I}\left[\exists m \in M : (|m - m'| \leq \epsilon \max(m, 200)) \wedge (|A(m) \wedge A(m')| > 0)\right]}{\sum_{m' \in M'} \mathbb{I}\left[|A(m')| > 0\right]} \tag{33}$$

$$\mathrm{AWP} = \frac{\sum_{m' \in M'} P_\theta(m')\mathbb{I}\left[\exists m \in M : (|m - m'| \leq \epsilon \max(m, 200)) \wedge (|A(m) \wedge A(m')| > 0)\right]}{\sum_{m' \in M'} P_\theta(m')\mathbb{I}\left[|A(m')| > 0\right]} \tag{34}$$

$$\mathrm{AF1} = \frac{2}{\mathrm{AR}^{-1} + \mathrm{AP}^{-1}} \tag{35}$$

$$\mathrm{AWF1} = \frac{2}{\mathrm{AWR}^{-1} + \mathrm{AWP}^{-1}} \tag{36}$$

Note that with FraGNNet and GrAFF-MS, the formula annotations for a given input molecule are deterministically computed: in the former case they are derived from the fragmentation DAG $\mathcal{F}^d$, while in the latter they are given by a static formula library. In both cases, the model's learned parameters only affect predicted peak intensities, not formula annotations. In contrast, ICEBERG uses a learned model to predict both formula annotations and peak intensities. This explains why seed variance for FraGNNet and GrAFF-MS is only reported in Table 3 for metrics that involve predicted intensities (AWP, AWF1).

In the case of the ICEBERG model, the metrics AWP or AWF1 are not well defined due to their dependence on predicted peak intensities. Unlike the other models, ICEBERG peak intensities $P_\theta(m')$ are only calculated after a 0.01 Da mass binning. This makes it impossible to evaluate $P_\theta(m'_1)$ and $P_\theta(m'_2)$ for predicted peaks $m'_1$ and $m'_2$ that end up in the same mass bin. For ICEBERG, the other metrics in Table 3 are calculated using the unbinned formula masses, *i.e.*, $m'_i = \mathrm{mass}(f^i)$ for each predicted formula $f^i$.

### A.16  Ensembling

Let $\left\{\theta^k : \theta^k \sim P(\theta)\right\}_{k=1}^K$ denote the set of parameters for an ensemble of $K$ FraGNNet-D4 models, sampled IID from a distribution over parameters $P(\theta)$. We perform ensembling in the latent space, using Equations 37 and 38 to calculate the ensemble latent distributions $P_\theta^K(n|f)$ and $P_\theta^K(f)$ (respectively):

$$P_\theta^K(n|f) = \frac{1}{K} \sum_k P_\theta^k(n|f) \approx \bar{P}(n|f) = \mathbb{E}_{\theta \sim P(\theta)}\left[P_\theta(n|f)\right] \tag{37}$$

$$P_\theta^K(f) = \frac{1}{K} \sum_k P_\theta^k(f) \approx \bar{P}(f) = \mathbb{E}_{\theta \sim P(\theta)}\left[P_\theta(f)\right] \tag{38}$$

The spectrum predictions are defined to be the average of the individual model predictions (Equation 39):

$$P_\theta^K(m) = \frac{1}{K} \sum_k P_\theta^k(m) \approx \bar{P}(m) = \mathbb{E}_{\theta \sim P(\theta)}\left[P_\theta(m)\right] \tag{39}$$

By defining the ensemble this way, we are implicitly assuming that $P_\theta^K(n|f)P_\theta^K(f) \approx P_\theta^K(n, f)$. Empirically we found that this was the case, as ensembling the joint distribution produced similar results.

Let $\mathbb{H}_\theta^K(N|f)$ denote the entropy of $P_\theta^K(n|f)$. Note that $\mathbb{H}_\theta^K(N|f)$ is a Monte Carlo approximation of $\bar{\mathbb{H}}(N|f) = \mathbb{E}_{n \sim \bar{P}(n|f)}\left[-\log \bar{P}(n|f)\right]$.

We want to show that ensembling increases latent conditional entropy. This is formalized in Equation 40:

$$\mathbb{E}_{\theta \sim P(\theta)}\left[\mathbb{H}_\theta(N|f)\right] \leq \bar{\mathbb{H}}(N|f) \tag{40}$$

*Proof.* Proving Equation 40 simply requires noting that entropy is strictly concave with respect to the probability density function. To make this clear, we introduce the notation $h(P(x)) = \mathbb{H}(X)$ where $h$ is the entropy function applied to the distribution $P(x)$.

$$
\begin{aligned}
\bar{\mathbb{H}}(N|f) &= h(\bar{P}(n|f)) \\
&= h(\mathbb{E}_{\theta \sim P(\theta)}\left[P_\theta(n|f)\right)] \\
&\geq \mathbb{E}_{\theta \sim P(\theta)}\left[h(P_\theta(n|f))\right] \\
&= \mathbb{E}_{\theta \sim P(\theta)}\left[\mathbb{H}_\theta(N|f)\right]
\end{aligned}
$$

$\square$

By the same reasoning we can get bounds for the isomorphic distributions (Equation 41):

$$\mathbb{E}_{\theta \sim P(\theta)}\left[\mathbb{H}_\theta(\tilde{N}|f)\right] \leq \bar{\mathbb{H}}(\tilde{N}|f) \tag{41}$$

Since normalized entropy is simply a scaled version of entropy, the inequalities in Equations 40 and 41 also hold if each entropy is replaced with its normalized counterpart.

### A.17  Isomorphic Fragment Annotation

For a given pair of ensembles $P_{\theta_1}^K$ and $P_{\theta_2}^K$, pairwise fragment annotation agreement (PFA) measures the average amount of mode agreement betwen their fragment annotation distributions $P_{\theta_1}^K(n|f)$ and $P_{\theta_1}^K(n|f)$.

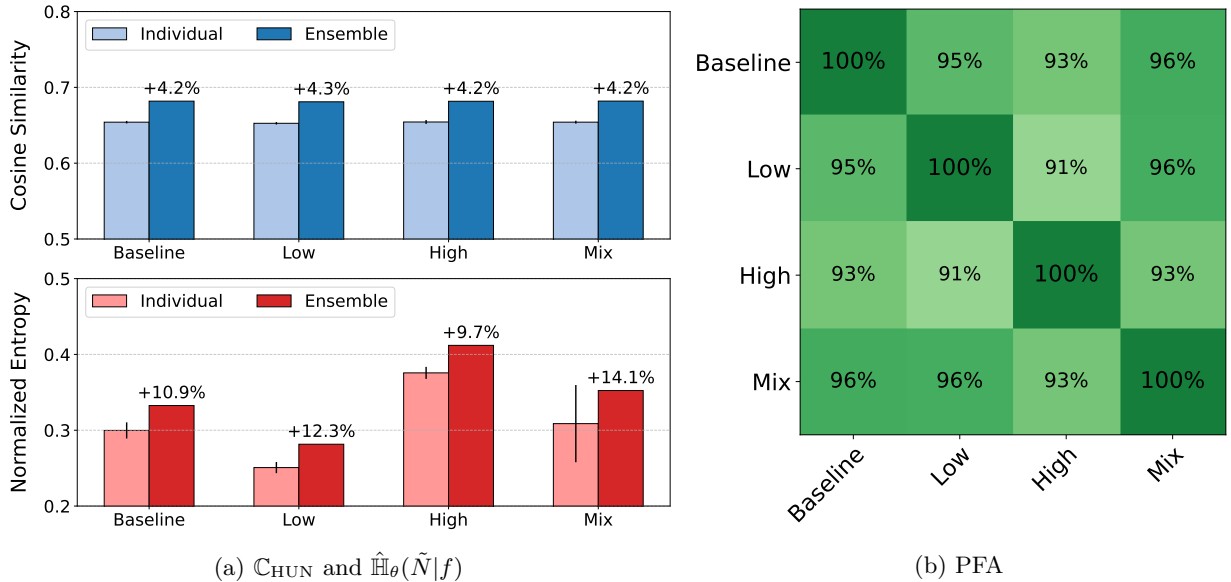

(a) $\mathbb{C}_{\text{HUN}}$ and $\hat{\mathbb{H}}_\theta(\tilde{N}|f)$

(b) PFA

Figure 6: (a) For each ensemble configuration (Baseline, Low, High, Mix), the cosine similarity $\mathbb{C}_{\text{HUN}}$ and normalized entropy of the annotation distribution $\hat{\mathbb{H}}_\theta(\tilde{N}|f)$ are reported. For both metrics, the average score of the individual models (Individual) is compared with the score of the ensemble (Ensemble). Each ensemble consists of $K = 15$ models. Standard deviations for the $K$ individual models are plotted as error bars. (b) Pairwise fragment annotation agreement (PFA) for all ensemble combinations are plotted in a matrix. The consensus agreement (CFA) is $\approx 89\%$.

More formally, let $F_i$ be the set of formula annotations in a predicted MS/MS spectrum $i$. PFA is defined using Equation 42.

$$\text{PFA} = \frac{1}{I} \sum_i \frac{\sum_{f \in F_i} \mathbb{I}\left[\arg\max_{\tilde{n}} P_{\theta_1}^K(\tilde{n}|f) = \arg\max_{\tilde{n}} P_{\theta_2}^K(\tilde{n}|f)\right]}{|F_i|} \tag{42}$$

Consensus fragment annotation agreement (CFA, Equation 43) is similar to PFA but measures agreement across all ensembles.

$$\text{CFA} = \frac{1}{I} \sum_i \frac{\sum_{f \in F_i} \mathbb{I}\left[\bigwedge_{a,b}\left(\arg\max_{\tilde{n}} P_{\theta_a}^K(\tilde{n}|f) = \arg\max_{\tilde{n}} P_{\theta_b}^K(\tilde{n}|f)\right)\right]}{|F_i|} \tag{43}$$

Figure 6 presents the same experiments as those covered by Figure 4, but focuses on the isomorphic annotation distribution $P_\theta(\tilde{n}|f)$ instead of $P_\theta(n|f)$. The general trends are the same: ensembling increases both cosine similarity and normalized entropy, PFA is $\geq 91\%$, and CFA is $\approx 89\%$. Note that for each ensemble configuration, $H_\theta(\tilde{N}|f) < H_\theta(N|f)$; this makes sense, as one would expect $P_\theta(\tilde{n}|f)$ to be more concentrated than $P_\theta(n|f)$ since the former does not distinguish between isomeric fragments.

Note that cases where only a single fragment is possible (*i.e.*, formulae $f \in F_i$ such that $P_\theta^K(n|f)$ is a degenerate distribution) are excluded from our analysis.

### A.18 Hyperparameter Optimization

For each model, we ran a random hyperparameter sweep with a budget of 100 samples on the InChIKey split, and selected the configuration with the best validation performance as measured by binned cosine similarity $\mathbb{C}_{\text{BIN}}$. The specific parameters that were optimized varied depending on the type of model.

### A.19 Inference Speed

We performed an inference speed comparison for FraGNNet and the baseline models using a random 1000-spectrum sample from the NIST20 Test split (Table 12). All timing experiments were conducted on a single node with modest compute (24-core Intel i9 13900K, 1 NVIDIA RTX A6000 GPU, 64 GB RAM). As expected, the models that do not rely on fragmentation (NEIMS, MassFormer, GrAFF-MS) were significantly faster than those that do (FraGNNet and ICEBERG). Fragment generation was performed on the CPU for both FraGNNet and ICEBERG (using 24 cores). Although GPU-based fragment generation is technically possible for ICEBERG, it is inefficient due to CPU-bound operations. For spectrum prediction, we used the GPU and tried batch sizes in $\{1, 4, 16, 32, 64, 128\}$, reporting only the fastest.

The inference times for the baseline models may be different from previously reported benchmarks. Our code uses sparse spectrum batching for both training and inference: this approach improves memory efficiency for models like FraGNNet that predict a highly variable number of peaks, since no peak padding is required. However, sparse batching can slow down inference, particularly for models like GrAFF-MS that always output a large fixed number of peaks. The authors of GrAFF-MS reported an average runtime of 3 ms per spectrum on the NIST20 dataset when using a single GPU (Murphy et al., 2023): this is roughly 30x faster than the time reported in our experiment. The authors of ICEBERG reported a runtime of approximately 0.86 s per spectrum on a single CPU core (Goldman et al., 2024): this is in line with our reported time of 1.62 s, although our experiment performs intensity prediction (ICEBERG-Score) on the GPU.

Table 12: Inference Speed. FRAGMENT is the time taken to generate fragments (where applicable); SPECTRUM is the time taken to predict the spectrum (excluding the FRAGMENT time); COMBINED accounts for both. TOTAL refers to the time taken to predict 1000 random spectra from the NIST20 Test split, while SINGLE refers to the average time for one spectrum (per-core for FRAGMENT times, since fragment generation occurs on a 24-core CPU). All times are in seconds.

| MODEL | FRAGMENT | | SPECTRUM | | COMBINED | |
|---|---|---|---|---|---|---|
| | TOTAL | SINGLE | TOTAL | SINGLE | TOTAL | SINGLE |
| FRAGNNET-D4 | 63.902 | 1.534 | 4.223 | 0.004 | 68.125 | 1.538 |
| FRAGNNET-D3 | 14.289 | 0.343 | 2.195 | 0.002 | 16.484 | 0.345 |
| ICEBERG | 66.723 | 1.601 | 21.662 | 0.022 | 88.385 | 1.623 |
| MASSFORMER | 0.000 | 0.000 | 2.991 | 0.003 | 2.991 | 0.003 |
| NEIMS | 0.000 | 0.000 | 1.884 | 0.002 | 1.884 | 0.002 |
| GRAFF-MS | 0.000 | 0.000 | 57.047 | 0.057 | 57.047 | 0.057 |

