# OpenReview forum: "FraGNNet: A Deep Probabilistic Model for Tandem Mass Spectrum Prediction"
_TMLR — Accepted by TMLR_

### Review · Reviewer_NwZY · 2025-06-06

**Summary Of Contributions:**

"FraGNNet: A Deep Probabilistic Model for Tandem Mass Spectrum Prediction" presents a method solve compound to MS/MS task using a graph neural networks-based probabilistic model. In the manuscript, the authors present two contributions to the field: (1) an efficient method to generate fragmentations graphs for a compounds, and (2) a method to predict the intensity of those fragments as a probabilistic model.The fragmentation graph approach, although not entirely novel, takes a clever approach to inject expert knowledge into this challenging task. Furthermore, the formulation of the spectrum prediction task as a probabilistic one clearly reflects what is known about the data collection process and demonstrates good performance.

**Audience:**

Yes

**Broader Impact Concerns:**

I thoroughly enjoyed reading this manuscript and think it will be impactful for the field.

**Claims And Evidence:**

Yes

**Requested Changes:**

1. Present a more comprehensive collection of metrics for the retrieval task.

2. Although the *MassSpecGym* paper (https://arxiv.org/abs/2410.23326) is cited, this manuscript does not present benchmark results for it. I recommend including benchmark results for retrieval in this benchmark dataset for others to compare against in the future.

**Strengths And Weaknesses:**

I found this manuscript to well-written with demonstrated expertise in both the mass spectrometry and modeling domains. I believe the primary strength of this manuscript is that the authors were able to combine these expertise to truly devise a modeling approach informed by a domain knowledge. Unlike some field, the data available to train such models is limited; thus, the described fragmentation graph approach and details such as not forcing the model to learn how to calculate m/z values, make the approach data and parameter-efficient.
A particular detail that I found clever was the approach the handling of "Out-of-Support" m/z predictions and their treatment in the loss function.

Although I think the presented approach is sound, my main critique is that the performance improvement over SOTA is sometimes modest, particularly for the retrieval task. This is likely the most important task presented in the experiment results, as it most closely reflects the primary way that such a model would be used by practitioners. Furthermore, the metrics presented for the retrieval task are limited to top-k accuracy. Here, top-k accuracy seems like a odd choice as apposed to top-k recall or precision. That being said, the retrieval task is difficult and even modest improvement can push the field forward.

---

> ### Author Response · Authors · 2025-07-10
> **Response to reviewer NwZY**
>
> We thank the reviewer for the positive and insightful feedback. We provide pointwise responses to their critiques below:
>
> > Present a more comprehensive collection of metrics for the retrieval task
>
> We agree that additional metrics would be insightful, and will add mean reciprocal rank to the final paper. Please see our [response to reviewer abgR](https://openreview.net/forum?id=UsqeHx9Mbx&noteId=wCl6pboXAv) regarding additional retrieval evaluations for FraGNNet and ICEBERG
>
> > Here, top-k accuracy seems like a odd choice as apposed to top-k recall or precision.
>
> In each retrieval task there is only a single positive reference (i.e. only one true structure match), all other references are negative. We are following the terminology used in [Goldman et al 2024](https://pubs.acs.org/doi/10.1021/acs.analchem.3c04654): in this case top-k accuracy is mathematically equivalent to top-k recall. Since there is only a single positive, for a given query and choice of k the top-k recall is always binary: 1 if the positive reference was ranked in the top-k and 0 otherwise. Similarly, top-k precision would reduce to top-k accuracy (i.e. recall) divided by k and therefore would not convey additional information.
>
> > Although the MassSpecGym paper (https://arxiv.org/abs/2410.23326) is cited, this manuscript does not present benchmark results for it.
>
> Our work is included in the MassSpecGym benchmark paper itself, and was the best-performing C2MS model in the original benchmark (Table 4: 0.52 cosine similarity, 46.64 top-1 accuracy).
>
> We note that the setup for this task differs from the experiments in our paper in the following ways:
> - In MassSpecGym, the spectra are not merged across collision energies (in our work, they are)
> - In MassSpecGym, spectra from both Orbitrap and Q-TOF instruments are included (in our work only Orbitrap spectra are included)
> - In MassSpecGym, the candidate set is larger with up to 256 molecules per spectrum and was not based on querying PubChem (in our work, we use 50 candidates per spectrum and perform candidate selection by querying PubChem for the 49 most similar molecules to the query)

---

> > ### Comment · Reviewer_NwZY · 2025-07-18
> >
> > Thank you for your responses. I have no further questions.

---

### Review · Reviewer_7cb5 · 2025-06-25

**Summary Of Contributions:**

The paper introduces FraGNNet, a model for compound to tandem mass spectrum prediction with a novel and efficient approach for fragment generation, and a probabilistic formulation for fragment intensity prediction. In addition, entropy estimation and ensembles are used regularize and interpret the model. FraGNNet achieves state-of-the-art accuracy in intensity prediction and a downstream compound retrieval task.

**Audience:**

Yes

**Claims And Evidence:**

Yes

**Requested Changes:**

- The claim for faster inference would benefit from a empirical experiment, measuring spectrum prediction speed comparing FraGNNet-D4, Iceberg and GrAFF-MS. Preferably, the experiment would differentiate the fragment generation, and intensity prediction when possible. The model parameter counts in the appendix suggest that FraGNNet-D4 should perform well.
- A heuristic is used to simplify the fragmentation graph. Would it be possible to evaluate the impact of this heuristic on model performance, similar to the evaluation of ICEBERG (+OptFrag)?

**Strengths And Weaknesses:**

Strengths:
+ The paper is clear and well written, and gives an in-depth overview of preliminary work.
+ The problem formulation is exhaustive and uses clear to follow math.
+ The experiements and comparions are well-designed and broad.
+ Latent entropy regularization and ensemble methods were used in an elegant way to asses sources for model error and to reduce them

Weakness:
- The paper mentions scaling issues of other models but does not provide benchmarks or comparisons

---

> ### Author Response · Authors · 2025-07-10
> **Response to reviewer 7cb5**
>
> We thank the reviewer for their thoughtful and positive evaluation of our work and for the constructive feedback provided. We provide pointwise responses to their critiques below:
>
> > The claim for faster inference would benefit from an empirical experiment, measuring spectrum prediction speed comparing FraGNNet-D4, Iceberg and GrAFF-MS. Preferably, the experiment would differentiate the fragment generation, and intensity prediction when possible. The model parameter counts in the appendix suggest that FraGNNet-D4 should perform well.
>
> We agree that timing experiments are important. In our [response to reviewer abgR](https://openreview.net/forum?id=UsqeHx9Mbx&noteId=wCl6pboXAv), we have presented some results for FraGNNet and ICEBERG (the two slowest but best-performing models). We will include results for all models in the final paper. We know from previous work ([Goldman et al 2024](https://pubs.acs.org/doi/10.1021/acs.analchem.3c04654)) that MassFormer and NEIMS should be much faster. We would note that for fragmentation models like FraGNNet and ICEBERG, the parameter count is not the main contributor to the runtime, rather the fragment generation (which relies on either combinatorial bond removal or autoregressive generation).
>
> > A heuristic is used to simplify the fragmentation graph. Would it be possible to evaluate the impact of this heuristic on model performance, similar to the evaluation of ICEBERG (+OptFrag)?
>
> We believe that implementing a version of FraGNNet that uses the full Directed Acyclic Graph (DAG) would be challenging.
>
> Our existing fragmentation algorithms relies on two key assumptions:
> - Assumption #1: there is a maximum number of bond breakages $j$ to reach any given fragment from the precursor molecule
> - Assumption #2: the bond set $E$ does not include bonds involving hydrogens, and does not treat higher order bonds differently from single bonds
>
> The approximate fragmentation algorithm has a worst-case performance that scales $\Theta(|E|^{j+1})$, where $j = 4$ for the D4 case.
>
> Relaxing Assumption #1 would correspond to increasing $j$ (up to a maximum of $j = |E|$), If we increased it up to $j = 5$, in the worst case this would increase runtime by a factor of $|E|$, which in our dataset is 20 on average and at most 68.
>
> Relaxing Assumption #2 by including bonds that involve hydrogens would increase the number of edges $|E|$ by a factor of 1.79 on average and 3.60 at most (in our dataset). In the worst case, this could increase runtime by a factor of roughly $3.60^{4+1} = 604$. This additional complexity would only account for hydrogen removals - accounting for transfers (i.e. movements of hydrogens from one part of the molecule to another) would be even more expensive.
>
> We already chose the largest value of $j$ such that the algorithm was practical to run on the molecules in our dataset. Consider that with D4, the largest DAG has ~30,000 fragment nodes (Table 3 in the paper).
>
> The difficulty is not only in computing the DAG, but also running the model on such a large set of fragments. VRAM is of particular concern: even with D4 we relied on a dynamic batching algorithm to efficiently train on GPUs while avoiding memory errors.
>
> Finally, we would like to provide a clarification on the ICEBERG (+OptFrag) ablation. Both ICEBERG and FraGNNet make assumptions that simplify the fragmentation DAG. In ICEBERG’s case, the DAG is approximated as a tree. Similar to FraGNNet, the explicit placement of hydrogens and higher-order bonds are not considered during fragmentation, and there is a limit on the maximum number of bond breakages that can occur.
>
> The ICEBERG (+OptFrag) ablation is not testing those simplifying assumptions, rather it is testing the effect of replacing the generative model (ICEBERG-GEN) that is used to predict the simplified DAG at inference time with the simplified DAG itself. Because FraGNNet does not have a generative model for DAG prediction (i.e. it always uses the simplified DAG), it is not possible to have an equivalent ablation.

---

> > ### Comment · Reviewer_7cb5 · 2025-07-18
> >
> > Thank you for the inference time benchmarks and the extended explanation about the impact of heuristics in the fragmentation algorithm and clarification about the OptFrag ablation. Your theoretical estimates about relaxing the two assumptions give a good intuition. I have no further questions.

---

### Review · Reviewer_abgR · 2025-06-30

**Summary Of Contributions:**

This paper introduces FraGNNet, a molecule-to-mass spectrum forward simulator that combines combinatorial fragmentation with graph neural networks (GNNs). Building upon the ICEBERG framework, FraGNNet replaces the learned fragmentation model with a deterministic combinatorial enumeration strategy, similar to those used in earlier non-learning-based methods such as MetFrag. The authors further update the neural network architecture to accommodate this new design. Experimental results demonstrate that this approach leads to improved performance in both spectrum generation and molecular structure retrieval tasks.

**Audience:**

Yes

**Broader Impact Concerns:**

No broader impact concerns from the reviewer.

**Claims And Evidence:**

Yes

**Requested Changes:**

1. **Clarify Baseline Discrepancies** (Critical):
   Please elaborate on the differences in evaluation setup (especially for MassFormer and ICEBERG) compared to those reported in [arXiv:2502.17874v1](https://arxiv.org/pdf/2502.17874v1). If feasible, consider re-running the baselines under a standardized setting to ensure fair performance comparisons.

2. **Report Inference Time** (Important):
   Provide quantitative inference-time metrics for FraGNNet and compare them with peer methods. This would help assess the trade-off between performance gain and computational cost.

**Strengths And Weaknesses:**

## Strengths

* **Novel Integration of Combinatorial Fragmentation**: The key technical insight—replacing the learned fragment generator with combinatorial enumeration—is both elegant and impactful. It shows that a non-learned, deterministic strategy can be competitive or even superior in the molecule-to-spectrum (M2MS) generation task.
* **Thorough Empirical Evaluation**: The paper presents a comprehensive set of experiments to validate the design choices, including comparisons across multiple metrics and baselines.
* **Clarity and Presentation**: The paper is well-structured and clearly written, making it accessible to readers from both the cheminformatics and machine learning communities.

## Weaknesses

* **Baseline Reproducibility and Configuration Concerns**: A key concern lies in the reported performance of peer methods. As shown in recent literature (e.g., [arXiv:2502.17874v1](https://arxiv.org/pdf/2502.17874v1), Table 4 in the appendix), models such as **MassFormer** and **ICEBERG** have achieved higher top-1 retrieval accuracies than those reported in this paper. For example:

  * Top-1 accuracy for MassFormer: 0.252
  * Top-1 accuracy for ICEBERG: 0.251

    This paper reports lower values for both. While methodological differences (e.g., bin width of 0.01 in cosine similarity) may partially explain these discrepancies, retrieval accuracy is often used as a benchmark metric. The authors should clarify the setup and, if feasible, re-run baselines using configurations aligned with the literature for a fair comparison. It is worth noting that I do not mean to compare FraGNNet with any recent/concurrent methods, but want the ICEBERG and MassFormer baselines to be clarified.
* **Computational Efficiency**: The combinatorial fragmentation strategy, while effective, could introduce computational overhead. The paper does not report inference time or runtime comparisons against baseline models. This is important to assess the practical feasibility of FraGNNet.

---

> ### Author Response · Authors · 2025-07-10
> **Response to reviewer abgR (part 1)**
>
> We thank the reviewer for their assessment of our work and attention to detail regarding empirical comparisons. We provide pointwise responses to their critiques below:
>
> > Please elaborate on the differences in evaluation setup (especially for MassFormer and ICEBERG) compared to those reported in arXiv:2502.17874v1. If feasible, consider re-running the baselines under a standardized setting to ensure fair performance comparisons.
>
> Our experiments differ from those in [Wang et al 2025](https://arxiv.org/abs/2502.17874) in the following ways:
> - Although both papers use the NIST20 dataset (Orbitrap HCD [M+H]+ subset), the split proportions are different: our experiments use a 60/20/20 split, while their experiments use a 80/10/10 split.
> - The spectrum preprocessing in Wang et al 2025 involves taking the square root of the intensities. In our paper, we do not do this when evaluating similarity or retrieval. Furthermore, for the Wang et al 2025 evaluation, an m/z bin size of 0.1 Da was used, while we employ a bin size of 0.01 Da.
>
> There isn’t a consensus in the literature on how to set up retrieval evaluations, with variation in spectrum preprocessing and similarity metrics across different works ([Wang et al 2021](https://pubs.acs.org/doi/10.1021/acs.analchem.1c01465), [Young et al 2024](https://www.nature.com/articles/s42256-024-00816-8), [Goldman et al 2024](https://pubs.acs.org/doi/10.1021/acs.analchem.3c04654), [Bushuiev et al 2024](https://arxiv.org/abs/2410.23326), [Li et al 2021](https://www.nature.com/articles/s41592-021-01331-z)). We think the approach outlined in our paper is appropriate, and we went to great lengths to implement all baseline models in our framework to achieve a fair comparison.
>
> However, we think the reviewer touches on an important point - choices in spectrum preprocessing and similarity metrics can influence model performance. We decided to investigate these differences on the retrieval evaluation. Due to limited time for the rebuttal, we have prioritized the two most competitive models (FraGNNet-D4 and ICEBERG) on the more challenging Scaffold split. The results are presented in the table below: “cos” indicates cosine similarity; “js” indicates jensen-shannon similarity (also sometimes called spectral entropy, see [Li et al 2021](https://www.nature.com/articles/s41592-021-01331-z)); “sqrt” indicates square root transform on peak intensities; “0.01” indicates 0.01 Da binning; “hun” indicates hungarian matching (10 ppm).
>
> | model | ranking_method | top_1_ratio | top_3_ratio | top_5_ratio | top_10_ratio | mean_reciprocal_rank |
> |:------------|:-----------------|:----------------|:----------------|:----------------|:----------------|:-----------------------|
> | FraGNNet-D4 | cos_0.01 | 0.21 +/- 0.007 | 0.468 +/- 0.004 | 0.614 +/- 0.01 | 0.797 +/- 0.005 | 0.391 +/- 0.006 |
> | FraGNNet-D4 | cos_0.01_sqrt | 0.296 +/- 0.006 | 0.594 +/- 0.004 | 0.73 +/- 0.003 | 0.876 +/- 0.003 | 0.485 +/- 0.004 |
> | FraGNNet-D4 | cos_hun | 0.211 +/- 0.007 | 0.478 +/- 0.005 | 0.623 +/- 0.007 | 0.805 +/- 0.004 | 0.395 +/- 0.006 |
> | FraGNNet-D4 | cos_hun_sqrt | **0.303 +/- 0.006** | **0.598 +/- 0.003** | **0.732 +/- 0.004** | **0.876 +/- 0.003** | **0.489 +/- 0.004** |
> | FraGNNet-D4 | js_0.01 | 0.289 +/- 0.009 | 0.584 +/- 0.003 | 0.723 +/- 0.003 | 0.872 +/- 0.002 | 0.477 +/- 0.006 |
> | FraGNNet-D4 | js_0.01_sqrt | 0.297 +/- 0.008 | 0.594 +/- 0.004 | 0.73 +/- 0.004 | 0.869 +/- 0.004 | 0.484 +/- 0.005 |
> | FraGNNet-D4 | js_hun | 0.278 +/- 0.008 | 0.566 +/- 0.003 | 0.699 +/- 0.004 | 0.854 +/- 0.004 | 0.463 +/- 0.005 |
> | ICEBERG | cos_0.01 | 0.204 +/- 0.005 | 0.442 +/- 0.009 | 0.582 +/- 0.008 | 0.764 +/- 0.008 | 0.376 +/- 0.004 |
> | ICEBERG | cos_0.01_sqrt | 0.27 +/- 0.006 | 0.541 +/- 0.003 | 0.68 +/- 0.005 | 0.836 +/- 0.006 | 0.449 +/- 0.003 |
> | ICEBERG | js_0.01 | 0.264 +/- 0.007 | 0.529 +/- 0.004 | 0.666 +/- 0.005 | 0.829 +/- 0.006 | 0.442 +/- 0.004 |
> | ICEBERG | js_0.01_sqrt | 0.279 +/- 0.009 | 0.55 +/- 0.007 | 0.683 +/- 0.008 | 0.836 +/- 0.009 | 0.456 +/- 0.007 |
>
> As expected, retrieval performance for a given model varies considerably.  However, it is worth noting that for each metric, the best FraGNNet-D4 configuration outperformed the best ICEBERG configuration.
>
> Additionally, we note that results for FraGNNet-D3 are included in the MassSpecGym benchmark ([Bushuiev et al 2024](https://arxiv.org/abs/2410.23326)). MassSpecGym is a community effort to establish a standard method of evaluating models in terms of simulation and retrieval, facilitating model comparisons. For MassSpecGym, the retrieval and simulation evaluations are performed on unmerged spectra, using cosine similarity with 0.01 Da bin size and without intensity transformations.

---

> > ### Author Response · Authors · 2025-07-10
> > **Response to reviewer abgR (part 2)**
> >
> > > Provide quantitative inference-time metrics for FraGNNet and compare them with peer methods. This would help assess the trade-off between performance gain and computational cost.
> >
> > We agree that inference time experiments are important, and have included runtime experiments for ICEBERG and FraGNNet in the table below. We will include results for all models in the final version of the paper. Our experiments measure prediction time for 1000 random molecules from our NIST20 dataset, using the same 24-core node with a single A5000 GPU. We report times for CPU and GPU inference.
> >
> > | model | fragment_generation | intensity_prediction | total | per molecule |
> > |:--------------------------|:--------------------|:---------------------|:-------|:-------------|
> > | ICEBERG GEN-CPU INTEN-CPU | 64 | 53 | 117 | 0.117 |
> > | ICEBERG GEN-GPU INTEN-CPU | 129 | 53 | 182 | 0.182 |
> > | ICEBERG GEN-CPU INTEN-GPU | 64 | 21 | 85 | 0.085 |
> > | ICEBERG GEN-GPU INTEN-GPU | 192 | 21 | 213 | 0.213 |
> > | FraGNNet-D3 CPU | 14 | 15 | 29 | 0.029 |
> > | FraGNNet-D3 GPU | 14 | 2 | 16 | 0.016 |
> > | FraGNNet-D4 CPU | 64 | 31 | 95 | 0.095 |
> > | FraGNNet-D4 GPU | 64 | 4 | 68 | 0.068 |
> >
> > It seems that the average runtime of FraGNNet-D4 is comparable to that of ICEBERG. Additionally, our reported metrics for ICEBERG (0.085s per molecule, with CPU fragment generation and GPU intensity prediction) are roughly in line with what the ICEBERG authors reported in [Goldman et al 2024](https://pubs.acs.org/doi/10.1021/acs.analchem.3c04654) (~1s per molecule on a single core machine, which translates to ~0.04s per molecule with 24 cores). We also know from Goldman et al 2024 that NEIMS and MassFormer are much faster than ICEBERG (and by extension, FraGNNet).

---

> > > ### Comment · Reviewer_abgR · 2025-07-10
> > >
> > > Thank you for the response; I would like to see the updated results in future versions. I have no other questions

---

### Decision · Action_Editor_7ZiS · 2025-07-21

**Recommendation:** Accept as is

**Audience:**

Yes

**Audience Explanation:**

This is a strong paper tying together sound ML (entropy regularization and ensembling) to domain specific knowledge to innovate on SOTA GNN models.  Also, tandem mass spectrometry is an interesting comp bio application which has been gaining traction lately at major ML conference.  Readers may benefit from the clear description of this interesting application

**Claims And Evidence:**

Yes

**Claims Explanation:**

All reviewers agree that a major strength of the paper is the empirical verification of the proposed methods.